# Investigating the mediating role of emotional intelligence in the relationship between internet addiction and mental health among university students

**Girum Tareke Zewude**[1] *, **Derib Gosim**[1], **Seid Dawed**[1], **Tilaye Nega**[1], **Getachew Wassie Tessema**[2], **Amogne Asfaw Eshetu**[3]

1 Department of Psychology, Wollo University, Dessie, Ethiopia, 2 Department of Information System, Wollo University, Dessie, Ethiopia, 3 Department of Geography and Environmental Studies, Wollo University, Dessie, Ethiopia

* girum.tareke@wu.edu.et; girumtareke27@gmail.com

**Data Availability Statement:** All data are in the manuscript and/or supporting information files.

## Abstract

### Introduction

The widespread use of the internet has brought numerous benefits, but it has also raised concerns about its potential negative impact on mental health, particularly among university students. This study aims to investigate the relationship between internet addiction and mental health in university students, as well as explore the mediating effects of emotional intelligence in this relationship.

### Objective

The main objective of this study was to examine whether internet addiction (dimensions and total) negatively predicts the mental health of university students, with emotional intelligence acting as a mediator.

### Methods

To address this objective, a cross-sectional design with an inferential approach was employed. Data were collected using the Wong Law Emotional Intelligence Scale (WLEIS-S), Internet Addiction Scale (IAS), and Keyes' Mental Health Continuum-Short Form (MHC-SF). The total sample consisted of 850 students from two large public higher education institutions in Ethiopia, of which 334 (39.3%) were females and 516 (60.7%) were males, with a mean age of 22.32 (SD = 4.04). For the purpose of the study, the data were split into two randomly selected groups: sample 1 with 300 participants for psychometric testing purposes, and sample 2 with 550 participants for complex mediation purposes. Various analyses were conducted to achieve the stated objectives, including Cronbach's alpha and composite reliabilities, bivariate correlation, discriminant validity, common method biases, measurement invariance, and structural equation modeling (confirmatory factor analysis, path analysis, and mediation analysis). Confirmatory factor analysis was performed to assess the construct

**Funding:** The author(s) received no specific funding for this work.

**Competing interests:** The authors have declared that no competing interests exist.

validity of the WLEIS-S, IAS, and MHC-SF. Additionally, a mediating model was examined using structural equation modeling with the corrected biased bootstrap method.

## Results

The results revealed that internet addiction had a negative and direct effect on emotional intelligence (β = –0.180, 95%CI [–0.257, –0.103], p = 0.001) and mental health (β = –0.204, 95%CI [–0.273, –0.134], p = 0.001). Also, Internet Craving and Internet obsession negatively predicted EI (β = –0.324, 95%CI [–0.423, –0.224], p = 0.002) and MH (β = –0.167, 95%CI [–0.260, –0.069], p = 0.009), respectively. However, EI had a significant and positive direct effect on mental health (β = 0.494, 95%CI [0.390, 0.589], p = 0.001). Finally, EI fully mediated the relationship between internet addiction and mental health (β = –0.089, 95%CI [–0.136, –0.049], p = 0.001). Besides The study also confirmed that all the scales had strong internal consistency and good psychometric properties.

## Conclusion

This study contributes to a better understanding of the complex interplay between internet addiction, emotional intelligence, and mental health among university students. The findings highlight the detrimental effects of internet addiction on mental health, and the crucial mediating role of emotional intelligence.

## Recommendations

The findings discussed in relation to recent literature have practical implications for practitioners and researchers aiming to enhance mental health and reduce internet addiction among university students. Emotional intelligence can be utilized as a positive resource in interventions and programs targeting these issues.

### Author summary

This study examined the psychometric properties and relationships between internet addiction, emotional intelligence, and mental health among Ethiopian university students. The measures used - the Wong Law Emotional Intelligence Scale (WLEIS-S), Internet Addiction Scale (IAS), and Keyes' Mental Health Continuum-Short Form (MHC-SF) - were found to be reliable and valid in the Ethiopian context. The findings indicate that higher levels of internet addiction are associated with lower emotional intelligence and poorer mental health. Conversely, emotional intelligence was positively correlated with mental health. The study determined that internet addiction negatively predicts both emotional intelligence and mental health, while internet craving directly predicts emotional intelligence and indirectly and negatively predicts mental health through emotional intelligence. These results highlight the detrimental effects of excessive internet use on the mental well-being of university students. The authors recommend interventions to raise awareness, promote healthy digital habits, and enhance emotional intelligence as a means of supporting students' mental health and preventing internet addiction.

## 1. Introduction

Internet addiction is a growing behavioral issue characterized by compulsive and excessive use of the internet, which has been associated with various mental health problems, including psychological, emotional, and social well-being [1–2]. The widespread availability of the internet, coupled with the proliferation of smartphones and social media platforms, has contributed to the rise of internet addiction as a significant concern, particularly among adolescents [3–6]. The detrimental effects of excessive internet use on the well-being of young people have raised significant alarm [6–8].

Research has shown that internet addiction is linked to negative emotions, decreased quality of life, and overall mental health concerns, particularly among adolescents [9–12]. Moreover, it has been observed that emotional intelligence, the ability to recognize, understand, and manage emotions effectively, plays a crucial role in mediating the relationship between internet addiction and mental health [13–15]. Individuals with lower emotional intelligence may be more susceptible to internet addiction, while those with higher emotional intelligence are believed to be less prone to excessive internet use and its negative consequences [10,13–14,16].

Notably, most of the existing research on internet addiction, emotional intelligence, and mental health has focused on Western cultures, neglecting the cultural nuances that may influence these phenomena in diverse contexts [10–20]. Therefore, it is essential to conduct cross-cultural studies and develop culturally appropriate measures to understand the relationships between internet addiction, emotional intelligence, and mental health in specific cultural contexts, such as Ethiopia. By doing so, researchers can avoid the problem of assuming universal values and perspectives and tailor interventions and prevention strategies to address the needs and experiences of Ethiopian undergraduate university students.

Furthermore, there is a need to explore the mediating role of emotional intelligence between internet addiction and mental health, particularly among undergraduate university students in Ethiopia. While some studies have identified associations between internet addiction and emotional intelligence as well as internet addiction and mental health, the specific mechanisms and protective effects of emotional intelligence in this context remain unclear [1,4,6,11,21–26]. For example, a study conducted by researchers [6,12–14] found that internet addiction directly and indirectly influences adolescents' mental health and had a negative relationship with cognitive emotion regulation. Hence, it is preliminary examining these relationships and understanding the role of emotional intelligence as a mediator can contribute to the development of effective interventions that promote mental well-being and emotional understanding in the digital age.

Additionally, research should investigate the role of emotional intelligence across different age groups and professions to determine if its protective effects are consistent across diverse populations. This knowledge can inform the development of tailored interventions for specific groups. Moreover, there is a need for more research on effective intervention and prevention strategies for internet addiction and its impact on mental health [6,12]. Understanding the factors contributing to these problems can guide the development of evidence-based interventions that promote healthy online behaviors and good mental health based on the development of emotional intelligence [10–14,27–28]. Testing the psychometric suitability of the measures [29–33] and evaluating the effectiveness of these interventions in diverse populations and settings is crucial for their successful implementation [6,34–39]. Therefore, the present study aimed to examine the mediating role of EI in the relationship between IA and MH among undergraduate university students in Ethiopia.

## 1.1. Relationship between internet addiction and mental health

Research has consistently linked internet addiction to negative emotional outcomes and decreased quality of life, particularly among adolescents [6,9–14]. Importantly, emotional intelligence - the ability to recognize, understand, and manage emotions effectively - has been identified as a crucial protective factor against the detrimental effects of internet addiction on mental health [13–15]. Individuals with lower emotional intelligence may be more susceptible to developing internet addiction, while those with higher emotional intelligence are believed to be less prone to excessive internet use and its negative consequences [10,13–14,20]. However, the specific mechanisms by which emotional intelligence mediates the relationship between internet addiction and mental health remain unclear, especially in non-Western cultural contexts [6,10–20]. In addition, a study conducted by [6] found that internet addiction had a negative association with Internet craving, Addictive behavior, Internet obsession and Internet Compulsive Disorder with social well-being, emotional and psychological well-being. Examining these relationships in an Ethiopian sample of undergraduate university students can provide valuable cross-cultural insights and inform the development of tailored interventions to promote mental well-being and emotional understanding in the digital age.

## 1.2. The mediation role of emotional intelligence

The existing research has identified associations between internet addiction with emotional intelligence (EI) and internet addiction with mental health [1,4,11,21,24–28]. However, the literature has not adequately addressed the issue of the mediating role of emotional intelligence between internet addiction and mental health among undergraduate university students. Emotional intelligence is thought to play a significant role in mediating the relationship between internet addiction and mental health, particularly in adolescent populations [13,17]. Emerging evidence suggests that EI may play a crucial role in the relationship between internet addiction and mental health, and it may serve as a protective factor against the emotional burden experienced by different age groups and professions [18–19]. A study conducted by [6,27–28] found that internet addiction had a direct and indirect effect on mental health of adolescents. Furthermore, emotional intelligence can facilitate the development of strong interpersonal relationships, thereby reducing the need for excessive online socialization and acting as a protective factor against internet addiction [20]. Also, studies indicated that, individuals with higher levels of EI are better equipped to regulate their emotions, cope with stress, and maintain healthy interpersonal relationships, all of which are essential for good mental health [6,27–28]. In contrast, individuals with lower EI may be more vulnerable to the negative impacts of internet addiction on their mental well-being [6,14].

The cognitive-behavioral model of internet addiction [39] and the broaden-and-build theory of positive emotions [40] provide a theoretical framework for understanding the mediating role of EI. According to these theories, individuals with higher EI may be better able to manage their emotions and impulses, maintain balance in their digital and offline lives, and engage in activities that promote positive emotions and well-being, thereby mitigating the negative effects of internet addiction on mental health.

## 1.3. Gaps of the study and research hypotheses

Despite the extensive research on various aspects related to internet addiction [21–22], emotional intelligence [13–14,17], and mental health [6,24–25], there are still several significant research gaps that need to be addressed. Firstly, the specific mechanisms through which emotional intelligence acts as a protective factor against internet addiction and its impact on mental health require further exploration. Additionally, the mediating role of emotional

intelligence between internet addiction and mental health has not been well-documented, particularly from the perspective of developing countries. Second, research in the area of internet addiction and mental health has predominantly focused on Western cultures, and there is a pressing need for more studies that explore the impact of these phenomena in diverse cultural contexts. Cultural differences in values, norms, and technology use patterns can influence the relationship between internet addiction, social media overuse, and mental health outcomes. To tailor interventions and prevention strategies for specific populations, it is crucial to understand these cultural nuances [29–31].

Third, research should investigate the role of emotional intelligence in different age groups and professions to determine if its protective effects are consistent across diverse populations. This knowledge can inform the development of tailored interventions for specific groups. Finally, while some studies have explored the negative impact of internet addiction on mental health, there is a need for more research on effective intervention and prevention strategies using the cognitive-behavioral model of internet addiction [39] and the broaden-and-build theory of positive emotions [40]. By addressing these research gaps, we can gain a better understanding of the complex relationships between internet addiction, emotional intelligence, and mental health and this knowledge can guide the development of targeted interventions, policies, and educational programs to promote healthier and more balanced digital lifestyles [6,20,41–43].

Hence, based on the empirical and theoretical evidence mentioned above and the importance of the issue, this research aims to explore the predictive role of internet addiction on mental health through emotional intelligence. Additionally, the study investigates the mediating effects of emotional intelligence in this relationship. The following testable research hypotheses are proposed:

**RH1**: It is expected to have a significant relationship between Internet addiction, emotional intelligence, mental health, and demographic factors.

**RH2**: The Internet Addiction Scale (IAS), the Wong Law Emotional Intelligence Scale (WLEIS-S), and the Mental Health Continuum-Short Form (MHC-SF) exhibit high levels of reliability and validity.

**RH3:** Internet addiction has a negative and direct impact on emotional intelligence and mental health.

**RH4:** Emotional intelligence has a positive and direct impact on the mental health of university students.

**RH5:** Emotional intelligence mediates the relationship between internet addiction (dimensions and total) and mental health (refer to Figs 1 and 2).

## 2. Research method

### 2.1. Research design

The current study employed a quantitative research design with an associational and cross-sectional approach, deemed well-suited to achieve the stated objectives.

### 2.2. Study setting

The study's target population consists of undergraduate students enrolled in public universities in the Amhara Regional State of Ethiopia.

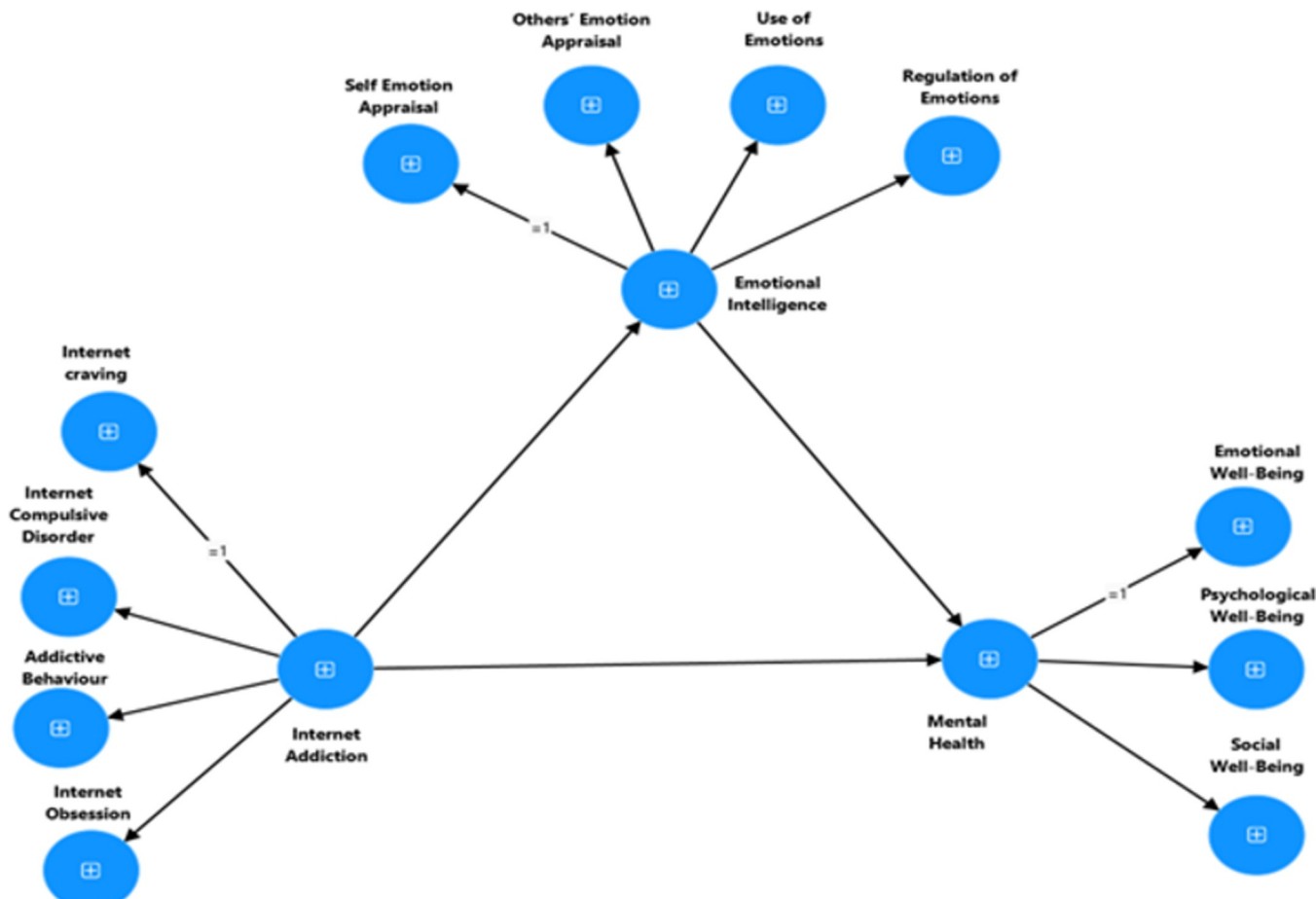

**Fig 1. Proposed mediation model to explain the association between Internet Addiction, Emotional Intelligence and Mental Health.**

## 2.3. Sample and sampling

The study was conducted in the Amhara Regional State of Ethiopia, focusing on two randomly selected public universities: Wollo University and Woldia University. The choice of this area was based on the researchers' extensive 16-year experience working in the region and the convenience of accessing accurate and efficient data due to the proximity of the study area to their workplace.

The sampling process involved dividing the sampling frame into subsections that represented characteristic groups. From each stratum, a random sample was selected. Initially, 889 university students were randomly chosen and invited to participate in the surveys. However, due to missing information, mistakes, or carelessness in data entry, 39 participants were excluded. This resulted in an effective response rate of 95.6%.

The final data consisted of 850 students, randomly split into two separate studies. The first study focused on validating the adapted instruments, while the second explored the mediation model through a complex structural equation modeling (SEM) analysis. In the first sample, there were 199 male and 101 female students. The second sample comprised 317 male and 233 female students. In total, the participants included 516 male students (60.7%) and 334 female students (39.3%). The mean age of the participants was 22.32 years, with a standard deviation of 4.04. To ensure comprehensive and reliable data, the researchers grouped the students'

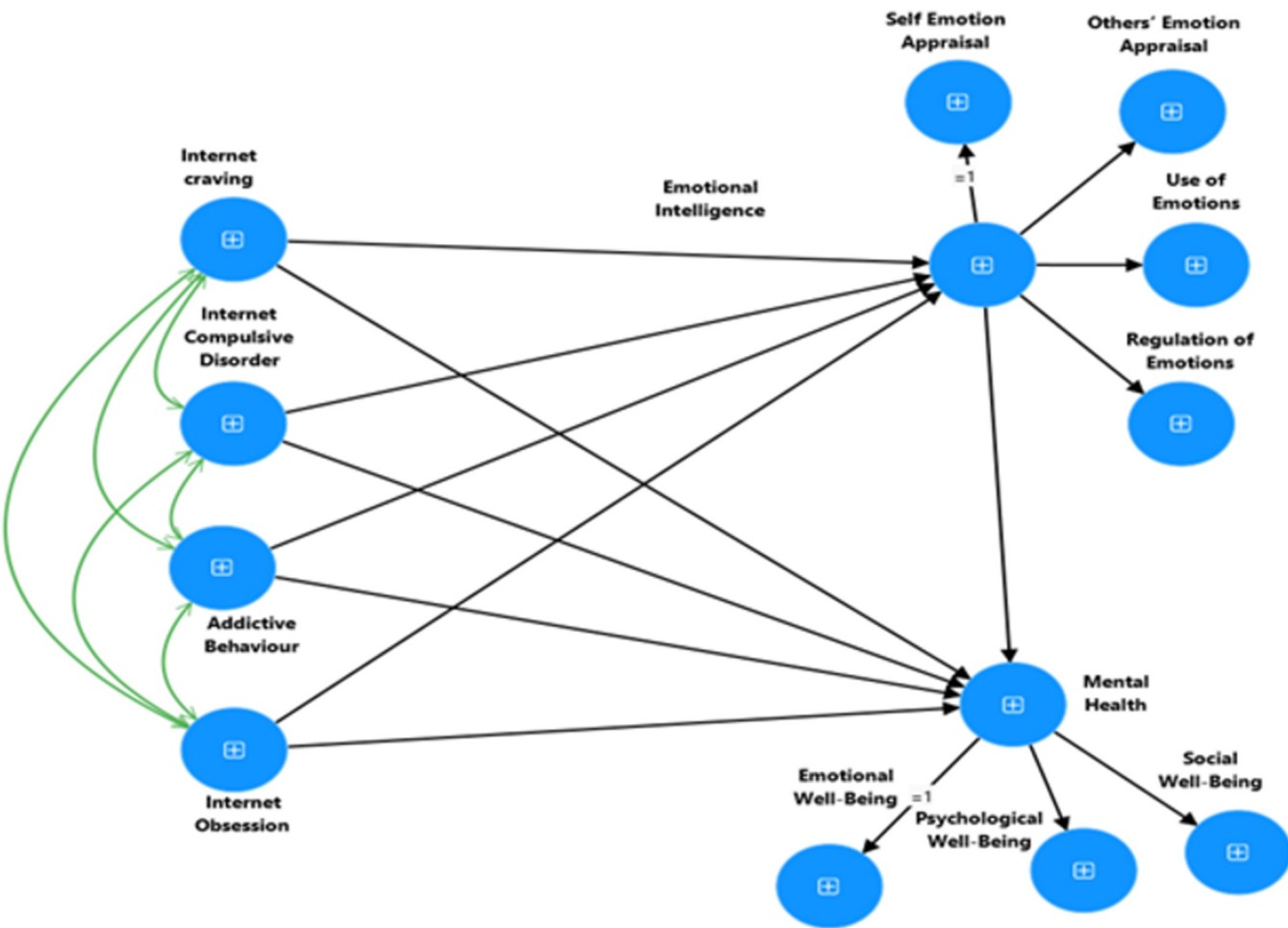

**Fig 2. Proposed mediation model to explain the association between dimensions of Internet Addiction, Emotional Intelligence and Mental Health.**

year/batch into three levels. Initially, the students were stratified by year and gender. From the total sample of 850 students, the following numbers were selected using a simple random sampling technique: (a) Freshman (1st year): 233 students (124 male and 109 female), (b) Sophomore (2nd year): 260 students (160 male and 100 female), and (c) Senior (>2nd year): 357 students (232 male and 125 female). The sample size was determined based on established guidelines. A small sample size is typically around 100, a medium size is approximately 150, and a large size is considered to be greater than 200. To ensure statistically stable estimates and minimize sampling errors, it is recommended to have a sample size of 200 or more [44]. Therefore, in line with these recommendations and considering the power of the test, the sample size in this study was determined.

## 2.4. Instruments

**2.4.1. Socio-demographic information.** Undergraduate students self-reported their gender, batch, age, and university.

**2.4.2. Internet Addiction Scale (IAS).** The IAS developed by [23] and later adapted in Ethiopian context by [6] used to assess an individual's excessive and compulsive internet use that interferes with daily functioning using a 17-item scale. The IAS was a seven-item response

option ranging from Very Strongly disagree (1) to Very Strongly agree (7). IAS has four main dimensions with very good psychometric properties: Internet craving (IC; Five Items; $\alpha$ = 0.771, CR = 0.836), Internet compulsive disorder (ICD; Four items; $\alpha$ = 0.776, CR = 0.857), Addictive behaviour (AB; Four Items; $\alpha$ = 0.741, CR = .802), Internet obsession (IO; Four Items; $\alpha$ = 0.663, CR = 0.862) and the total scale ($\alpha$ = 0.878). In this study, Cronbach's alpha coefficient, composite reliability (CR), construct validity, convergent and discriminant validity was acceptable based on global cut-off points/rule of thumb.

**2.4.3. Emotional Intelligence Scale (EIS-16).** EI was assessed using the 16-item EIS-16 [45] based on the Salovey Mayer EI framework [46]. Respondents rated each item on a 7-point Likert scale, ranging from 1 (very strongly disagree) to 7 (very strongly agree). EIS includes four main dimensions: Self-Emotion Appraisal (SEA); Others' Emotion Appraisal (OEA); Use of Emotions (UOE, and Regulation of Emotions (ROE), each of which are measured on four items [47]. The reliability and construct validity of the Amharic version tested in this study was acceptable (see details under the measurement model). In this study, the Cronbach's alpha, CR, and construct validity were found to be acceptable for the Ethiopian cultural context.

**2.4.4. Mental Health Continuum-Short Form [MHC-SF].** MHC-SF is the most widely used instrument designed to measure the status of mental health of an adolescent. The Keyes' Mental Health Continuum-Short Form [MHC-SF] was used to measure mental health of the participants [25]. The MHC-SF instrument was used to assess 'frequency of happiness, social belongingness to a community and managing the psychological functioning of the daily life [6,25]. Three major clusters: social, emotional, and psychological well-being were used to assess the healthy functioning of adolescents' mental health. Respondents The scale covered three dimensions: social well-being (5 items; $\alpha$ = 0.74), emotional well-being (3 items; $\alpha$ = 0.85), and psychological well-being (6 items; $\alpha$ = 0.84) and comprised 14 items [48]. The overall scale of MHC-SF reliability was $\alpha$ = 0.91. Respondents rate each item on a 7-point Likert scale, ranging from 1 (Very strongly disagree) to 7 (Very strongly agree). This scale possessed excellent construct validity and reliability [25]. Besides, the reliability and validity of the Amharic version have been proven in the Ethiopian context [6].

## 2.5. Statistical data analysis

IBM SPSS 29.0 and Smart-PLS 4.1.0.3 were used to perform the analyses. The psychometric properties and the complex mediation analyses were two essential aspects of this study. To test an instrument for psychometrically suitability, it is recommended to apply several methods and follow a scientific procedure in their assessment in study one followed by the complex mediation analysis using bootstrapping corrected bias. The main rationale of testing psychometric properties was because of cross-cultural validation is threatened by methodological difficulties, including those stemming from the translation of the questionnaire and the measurements of other instruments [33]. Therefore, in this study, validation was done following the guidelines proposed by [49]: (a) initial translation/forward translation, (b) translation synthesis, (c) back translation, (d) expert/translator review, and (e) administration and validation. In addition, the instruments were validated based on the recommendation of [30, 32]. Overall, the validation and the mediation findings were obtained through four processes.

i. **Multi-collinearity.** VIF and tolerance were used to identify multi-collinearity in statistical data, following the recommendations of [32, 50]. In addition, the Harman single-factor test was used to examine common method variance bias.

ii. **Evidence of reliability.** CR and the Cronbach's alpha coefficient were used to test the internal consistency of the subscales. Excellent internal consistency is shown by values over 0.90,

good internal consistency is indicated by values between 0.80 and 0.90, and acceptable internal consistency is demonstrated by values between 0.70 and 0.80 [47,51–53].

iii. **Confirmation of construct validity through convergent, divergent, and discriminant validity.** Average variance extracted and maximum shared variance were used to assess convergent and discriminant validity. AVE values greater than 0.5 are indicative of good convergent validity in a factor. Additionally, variables with a sufficient level of discriminant validity have an MSV value is lower than their AVE value [32].

iv. **Measurement invariance**. We used CFA to examine the psychometric equivalence of the variables across distinct groups for measurement invariance (MI) [29]. In this work, a single-group CFA and multi-group CFA with four MI phases were used in accordance with accepted scientific practices [29]. Stage 1 involved conducting a configural invariance to create a baseline model that could be used for all groups without restriction, with the tested construct being the same in each group. Stage 2 of the analysis looked at the metric measurement invariance (MMI), which observed how indicators were reacted to by various groups using the same constrained factorial loadings. Stage 3 involved scalar MI, often known as strong invariance (SMI). In this test, factor loadings and indicator intercepts were limited uniformly across groups. In the fourth stage, strict invariance (RMI), or residual measurement invariance (RMI), was assessed. RMI represents the similarity of metric and scalar invariant items' residuals [29]. Following [29,32]. Using multi-group CFA, the MI four sequential-staged analysis in the current study produced the following recommendation criteria. For metric, scalar, and residual invariance, the CFI and TLI ranged from 0 (perfect) to 0.01 (acceptable), and 0.015 (RMSEA) [29,31–32].

v. **Structural Equation Modeling (SEM).** This study investigated how well construct validity and causal relationship could explain the study variables [32]. Hence, SEM the best statistical Tanique to fit with the research purpose used to analyze the relationships between observed and latent (unobserved) variables [32]. We used three types of SEM in this study for testing causal relationship between the exogenous and endogenous variables of which there are three types: measurement model, structural model and path analysis [32]. ***Measurement Model*** is used to specifies how the observed variables are related to the underlying constructs they are intended to measure done through Confirmatory Factor Analysis (CFA). CFA tests a measurement theory by providing evidence of the validity of individual measures using the model's overall fitness and other evidence of construct validity [31–32,47]. B) ***Structural model*** is specifying the hypothesized causal relationships among the latent variables and how they influence each other; (C) ***path analysis*** used to test and estimate complex relationships among variables currently used in our study [32]. We evaluated the goodness-of-fit using normed chi-square ($\chi2/df$), the Tucker Lewis Index (TLI), Comparative Fit Index (CFI), and Root Mean Squared Error of Approximation (RMSEA). Measurement and structural models are generally deemed to exhibit excellent and sufficient fit if $\chi2/df$ is below 3 or 5, RMSEA and SRMR are below 0.08 and 0.01, and TLI and CFI are more than above 0.95 and 0.90, respectively [54]. The hypothesized model described in Figs 1 and 2 was examined using the 95% bias-corrected confidence intervals to examine indirect effects using the bootstrap method and 5000 resamples.

## 2.6. Procedures and informed consent statement

The questionnaire applied incorporated 49 questions, measuring the EI (16 questions), the Internet addiction Scale (17 questions), Mental health Continuum Short Form (14 questions)

and three socio-demographic factors. Paper and pencil were used by every participant to complete the surveys. The American Psychological Association's ethical guidelines and standards, the Wollo University, the Internal Review Board, and standard data collection process were all followed. Participation was voluntary, and the researchers assured the participants that their data would be anonymized. The 1964 Helsinki Declaration items 21 CFR 56 (Institutional Review Boards, IRB) and 21 CFR 50 (Protection of Human Subjects) were adhered in this study. This study received an ethical approval letter of IRB from Wollo University, Institute of Teachers Education and Behavioral Sciences (certificate number: Ref. 419/2023).

## 3. Results

### 3.1. Results of preliminary analysis

**3.1.1. Descriptive statistics, Skewness, and Kurtosis.** Table 1 presents an overview of the descriptive statistics, including means, standard deviations, skewness, and kurtosis, which serve as indicators of distribution normality [47,55–56]. In this study, data were considered to be normally distributed when skewness was within the range of -2 to 2 and kurtosis was within the range of -4 to 4 normality [47,52,55–56].

The results indicate that all the constructs in this study exhibited normal distribution. This is evident from the skewness values, which ranged from –.034 to –1.32, and the kurtosis scores, which ranged from .006 to 2.68. Additionally, the normality of distribution was further confirmed by examining the direct and indirect effects of the constructs, as depicted in Fig 3A–3D. Overall, the normal distribution of the data supports the validity and reliability of the statistical analyses conducted in this study.

**3.1.2. Multi-collinearity.** According to [32] if the tolerance values of predictor variables in a model are close to each other, it indicates the absence of an issue with multi-collinearity. Conversely, if the tolerance values are close to zero, it suggests the presence of multi-collinearity. The VIF statistic, which should ideally range from 0 to 5, with lower values being more desirable (approaching 0), is used to assess multi-collinearity. When the VIF score exceeds five, it indicates that certain predictor variables are linear combinations of others [32,47,52]. In our study, the VIF was below 5, and the tolerance limits for each independent variable were greater than or equal to 0.01 (see Table 2). Therefore, we concluded that the independent variables did not exhibit multi-collinearity issues based on the VIF and tolerance measures.

In addition, to assess the presence of common method bias in our study, we conducted the Harman single-factor test. The purpose of this test was to determine if a single dominant factor could explain the majority of the covariance among the measured variables, indicating the potential influence of common method bias. The results of the test revealed that all constructs exhibited a common method bias rate of 28.53%, which fell below the recommended fit requirements. Based on these findings, we concluded that the study findings were unlikely to be affected by bias resulting from common method variance. This suggests that the relationships and associations observed among the variables in our study are more likely to reflect genuine associations rather than being artifacts of methodological biases.

**3.1.3. Pearson correlation among the constructs.** Table 3 provides an overview of the interrelationships among the variables in the study. To test the first hypothesis and examine the associations between the independent factors and the dependent variable, correlation analysis was conducted following the guidelines outlined by [47,50–52]. The results indicated a negative correlation between IA with EIQ ($r = -0.167$, $p < 0.01$) and MH ($r = -0.269$, $p < 0.01$). Conversely, a positive correlation was observed between EI and MH ($r = 0.444$, $p < 0.01$). However demographic factors such as gender, age and batch/years of study had no correlation with the three main constructs (see Table 3 for detailed results).

**Table 1. Descriptive statistics, kurtosis and skewness(N = 300).**

| Variables | Minimum | Maximum | Mean | Std. Deviation | Skewness | Kurtosis |
|---|---|---|---|---|---|---|
| Internet craving | 5.00 | 25.00 | 19.7333 | 5.94 | −1.06 | 0.036 |
| Internet compulsive disorder | 4.00 | 20.00 | 14.8067 | 4.72 | −0.91 | −0.14 |
| Addictive behavior | 4.00 | 20.00 | 14.90 | 4.69 | −0.76 | −0.46 |
| Internet obsession | 4.00 | 20.00 | 15.57 | 4.79 | −1.09 | .026 |
| Internet Addiction | 17.00 | 85.00 | 65.01 | 17.32 | −1.14 | 0.84 |
| Self-Emotion Appraisal | 4.00 | 24.00 | 16.07 | 5.01 | −0.79 | .075 |
| Others' Emotion Appraisal | 4.00 | 24.00 | 15.46 | 5.06 | −0.64 | .006 |
| Use of Emotions | 4.00 | 24.00 | 14.80 | 5.21 | -0.34 | −0.60 |
| Regulation of Emotions | 4.00 | 22.00 | 13.11 | 4.54 | −0.38 | −0.56 |
| Emotional intelligence | 16.00 | 94.00 | 59.45 | 16.82 | −1.04 | 0.87 |
| Emotional Well-Being | 3.00 | 18.00 | 12.70 | 2.99 | −0.90 | 0.71 |
| Psychological Well-Being | 6.00 | 36.00 | 26.12 | 5.73 | −0.92 | 1.46 |
| Social Well-Being | 5.00 | 30.00 | 22.76 | 4.711 | −1.32 | 2.68 |
| Mental Health | 14.00 | 84.00 | 61.57 | 11.19 | −0.98 | 1.82 |

**3.1.4. Reliability and validity evidence of the main variables.** The first and most important task in cross cultural study was the measurement issues. To answer this issue the second research hypotheses was examined the construct validity, construct reliability, and internal consistency of the variables on undergraduate students in Ethiopian higher educational settings (see Table 4). Reliability scores above 0.90 indicate high reliability, scores between 0.80 and 0.90 suggest good reliability, and scores between 0.70 and 0.80 indicate adequate reliability [32,51–53]. The validity and reliability of the internet addiction, EIQ and mental health aspects were evaluated. The results for the reliability coefficients of the internet addiction scale (IAS) aspects were as follows: internet craving (α = 0.963, CR = 0.964), internet compulsive disorder (α = 0.950, CR = 0.951), addictive behavior (α = 0.945, CR = 0.944), and internet obsession (α = 0.950, CR = 0.951), all of which were excellent. The reliability of the WLEIS-S sub-dimensions were tested and the values for each of the WLEIS-S dimensions were as follows: Others' Emotion Appraisal (α = 0.941, CR = 0.944), Regulation of Emotions (α = 0.870, CR = 0.873), Self-Emotion Appraisal (α = 0.941; CR = 0.942), and Use of Emotions (α = 0.952; CR = 0.954). The reliability coefficients were also found for mental health (MH) dimensions as follows: emotional well-being (α = 0.861, CR = 0.862), psychological well-being (α = 0.929; CR = 0.929) and social well-being (α = 0.934, CR = 0.935). The study demonstrated the high reliability of the four sub-components of IA, the four dimensions of EIQ and the three key core elements of the MH construct in Ethiopian educational settings.

To assess the validity of these constructs, we examined their discriminant and convergent validity using the IAS, WLEIS-S, and MHC-SF. Table 4 presents the AVE, MVE, and CR values for the sub-components of the study variables. It was found that all four constructs of IA, the four EIQ elements, and the three dimensions of MH have good convergent validity (AVE > .05), indicating that the items are composed of fundamental components with reasonable correlation. Discriminant validity was evaluated by comparing the AVE values with the MSV values, and it was observed that the AVE values were higher than the MSV values, indicating acceptable discriminant validity. The AVE values for the sub-constructs of the IAS, WLEIS-S and the MHC-SF met the criteria for convergent and discriminant validity. Additionally, discriminant validity was assessed by comparing the AVE with squared inter-item correlations, and it was found that the AVE for all sub-constructs was higher than the squared correlation

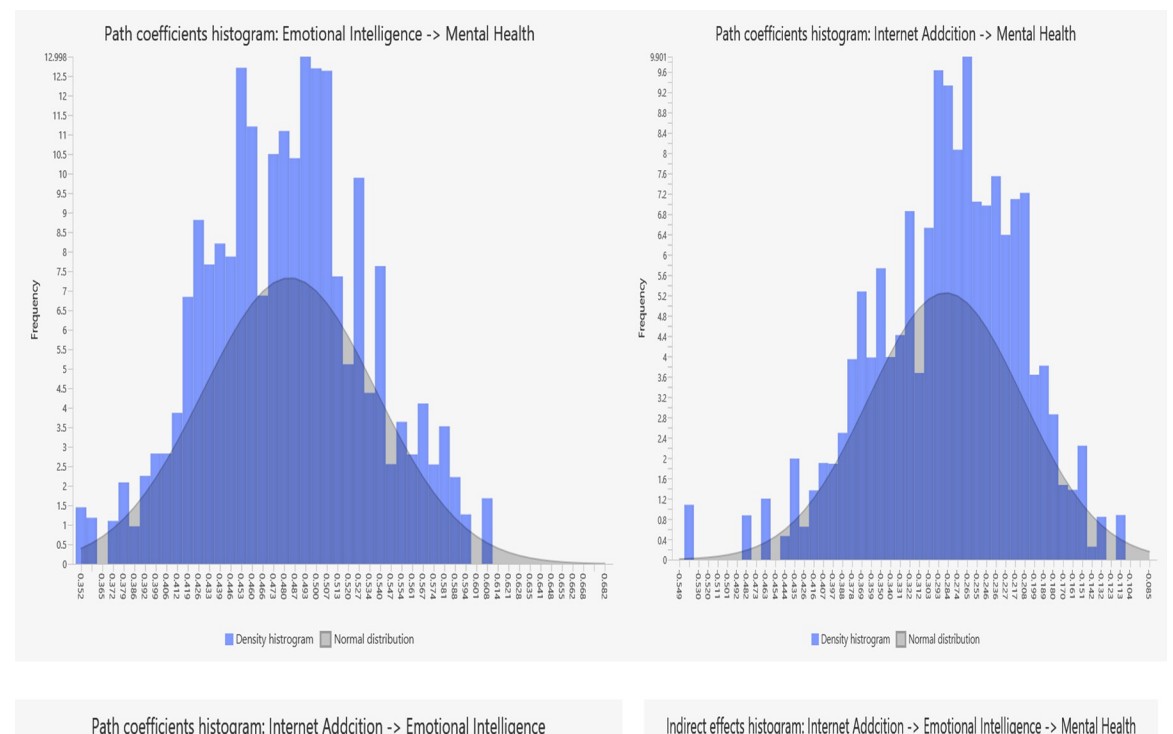

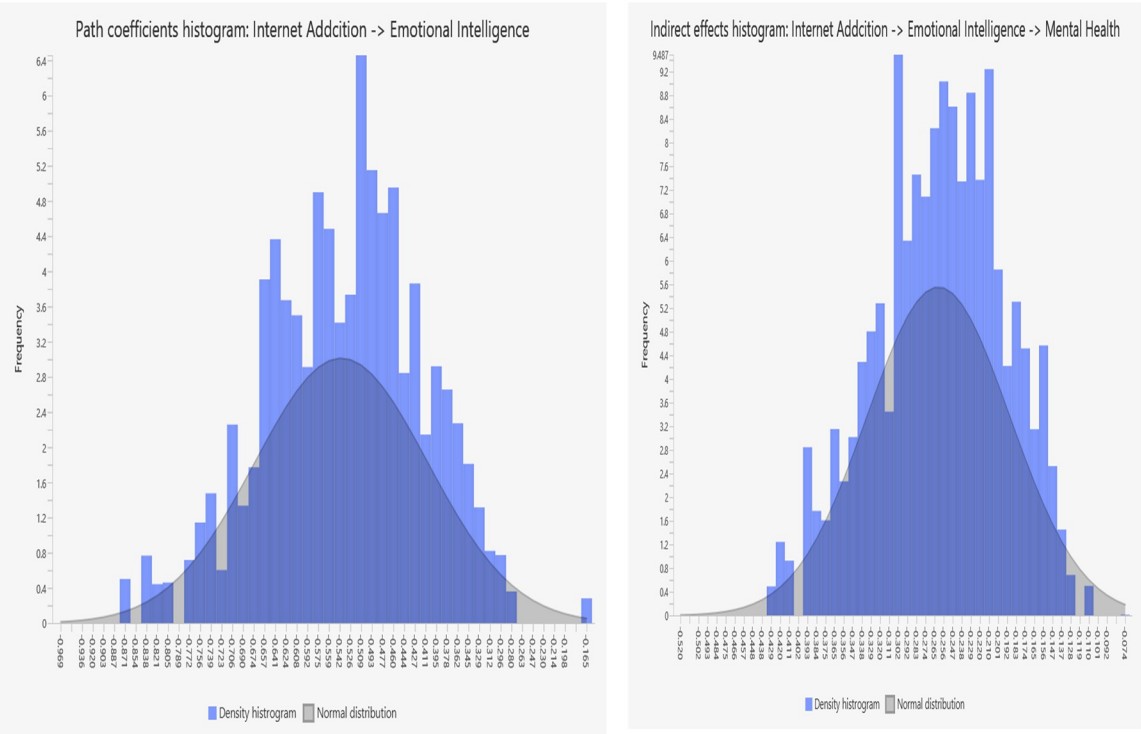

**Fig 3.** Normality distribution of: (Fig 3A) Emotional intelligence on Mental Health, (Fig 3B) Internet Addiction on Mental Health, (Fig 3C) Internet Addiction on Emotional Intelligence, and (Fig 3D) Internet Addiction on Emotional Intelligence and mental Health.

for each construct, indicating satisfactory discriminant validity. Overall, the IA, EIQ, and MH constructs meet the standards for convergent and discriminant validity in Ethiopian higher education and adolescent samples of university students.

**Table 2. Tolerance and VIF of multi-collinearity statistics of Internet addiction and Emotional Intelligence on Mental Health.**

| Model | Unstandardized Coefficients | Standardized Coefficients | t | Sig. | Collinearity Statistics | |
|---|---|---|---|---|---|---|
| | Beta | Beta | | | Tolerance | VIF |
| Internet addiction | −0.136 | −0. 122 | −4.004 | .000 | 0.970 | 1.031 |
| Emotional Intelligence | 0.376 | 0.444 | 14.522 | .000 | 0.970 | 1.031 |

**3.1.5. Measurement Invariance (MI).** As seen in Table 5, the four-step process of testing MI, more strict equality constraints were specified for model parameters between or among groups (for example, men vs. women; Freshman vs. Sophomore vs Senior) within a multiple-group CFA (MGCFA) following the guidelines of [29,31,47,52].

The configural model served as a starting point for subsequent tests and did not impose any equality constraints on parameters in the initial stage [29,31]. Configural invariance holds that comparable groups (same gender, and batch) should exhibit the same underlying factor structure. The metric model then looked at how similar the factor loadings were across groups for each item. Valid group comparisons require invariant factor loadings [29,31]. Following this, the scalar model looked for evidence of equal item intercepts, referring to the assessment whether mean differences at the item and factor levels can completely equal one another's variances. Finally, the rigorous model, or residual invariance, was used as the last step to determine whether the variances of each item's regression equations were equal across groups [29,31].

**Table 3. Pearson correlations (r) among the socio-demographic factors and the predictor variables with the Criterion variables (N = 550).**

| Variables | 1 | 2 | 3 | 4 | 5 | 6 | 7 | 8 | 9 | 10 | 11 | 12 | 13 | 14 | 15 | 16 | 17 |
|---|---|---|---|---|---|---|---|---|---|---|---|---|---|---|---|---|---|
| **1. sex** | 1 | | | | | | | | | | | | | | | | |
| **2. Age** | .070 | 1 | | | | | | | | | | | | | | | |
| **3. Batch** | −.013 | .006 | 1 | | | | | | | | | | | | | | |
| **4. IC** | −.044 | −.062 | −.076 | 1 | | | | | | | | | | | | | |
| **5. ICD** | .059 | −.059 | −.050 | 0.554** | 1 | | | | | | | | | | | | |
| **6. AB** | −.038 | -.016 | −.050 | 0.550** | 0.407** | 1 | | | | | | | | | | | |
| **7. IO** | .006 | .049 | .033 | 0.442** | 0.455** | 0.259** | 1 | | | | | | | | | | |
| **8. SEA** | −.052 | −.023 | .021 | −0.169** | −.094* | −.017 | −.046 | 1 | | | | | | | | | |
| **9. OEA** | −.071 | −.022 | −.034 | −.0209** | −.071 | −.049 | −.125** | 0.664** | 1 | | | | | | | | |
| **10. UOE** | .042 | .016 | −.011 | −.0120** | −.002 | −.020 | −.067 | 0.571** | 0.618** | 1 | | | | | | | |
| **11. ROE** | .081 | −.022 | -.027 | −0.173** | −0.139** | −.071 | −.162** | 0.365** | 0.347** | 0.310** | 1 | | | | | | |
| **12. EWB** | 0.197** | −.029 | -0.118** | −0.136** | −0.140** | −.034 | −.278** | 0.289** | 0.307** | 0.221** | 0.432** | 1 | | | | | |
| **13. PWB** | 0.239** | −.045 | -.077 | −0.210** | −0.124** | −.048 | −.313** | 0.250** | 0.324** | 0.218** | 0.294** | 0.608** | 1 | | | | |
| **14. SWB** | 0.144** | −0.100* | -0.114** | −0.232** | −0.182** | −.093* | −.288** | 0.389** | 0.408** | 0.237** | 0.341** | 0.627** | 0.711** | 1 | | | |
| **15. IA** | −.007 | −.030 | −.048 | 0.849** | 0.788** | 0.701** | .715** | −0.112** | −0.156** | −0.073 | -0.182** | −0.197** | −0.235** | −0.266** | 1 | | |
| **16. EI** | .018 | −−.016 | −.017 | −0.215** | −.097* | −.050 | −.128** | 0.829** | 0.845** | 0.805** | 0.644** | 0.399** | 0.348** | 0.439** | −0.167** | 1 | |
| **17. MH** | .022** | −.068 | −.013** | −0.227** | −0.168** | −.068 | −0.336** | 0.349** | 0.396** | 0.255** | 0.389** | 0.803** | 0.914** | 0.897** | −0.269** | 0.444** | 1 |

* & **. Correlation is significant at the 0.05 level and 0.01 level (2-tailed) respectively; AB = Addictive behavior, EI = Emotional intelligence, EWB = Emotional Well-Being, IA = Internet Addiction, ICD = Internet compulsive disorder, IC = Internet craving, IO = Internet obsession, MH = Mental Health, OEA = Others' Emotion Appraisal PWB = Psychological Well-Being, ROE = Regulation of Emotions, SEA = Self-Emotion Appraisal, SWB = Social Well-Being, UOE = Use of Emotions.

Table 4. Reliability and Validity Indices of the Study Variables (N = 300).

| | | | | | | | | |
|---|---|---|---|---|---|---|---|---|
| **Internet Addiciton Scale (IAS)** | | | | | | | | |
| **Models** | **α** | **CR** | **AVE (>.50\*)** | **MSV** | **Squared correlation** | | | |
| | (>.70\*) | | | | **IC** | **ICD** | **AD** | **IO** |
| Internet Craving (IC) | 0.963 | 0.964 | 0.841 | 0.68 | 1 | | | |
| Internet Compulsive Disorder(ICD) | 0.950 | 0.951 | 0.827 | 0.68 | 0.68 | 1 | | |
| Addictive Behaviour (AD) | 0.945 | 0.944 | 0.810 | 0.53 | 0.46 | 0.53 | 1 | |
| Internet Obsession(IO) | 0.950 | 0.951 | 0.826 | 0.42 | 0.37 | 0.37 | 0.42 | 1 |
| **Wong Law Emotional Intelligence Scale (WLEIS-S)** | | | | | | | | |
| **Models** | **α** | **CR** | **AVE (>.50\*)** | **MSV** | **Squared correlation** | | | |
| | (>.70\*) | | | | **SEA** | **OEA** | **UOE** | **ROE** |
| Self-Emotion Appraisal (SEA) | 0.941 | 0.942 | 0.803 | 0.66 | 1.00 | | | |
| Others' Emotion Appraisal (OEA) | 0.941 | 0.944 | 0.806 | 0.66 | 0.66\*\* | 1.00 | | |
| Use of Emotions (UOE) | 0.952 | 0.954 | 0.835 | 0.61 | 0.53\*\* | 0.61\*\* | 1.00 | |
| Regulation of Emotions (ROE) | 0.870 | 0.873 | 0.633 | 0.34 | 0.33\*\* | 0.34\*\* | 0.27\*\* | 1.00 |
| **Mental Health Continuum-Short Form (MHC-SF)** | | | | | | | | |
| **Models** | **α** | **CR** | **AVE (>.50\*)** | **MSV** | **Squared correlation** | | | |
| | (>.70\*) | | | | **EWB** | **PWB** | **SWB** | |
| Emotional Wel-Being(EWB) | 0.861 | 0.862 | 0.674 | 0.30 | 1.00 | | | |
| Psychological Well-Being(PWB) | 0.929 | 0.929 | 0.686 | 0.38 | 0.30\*\* | 1.00 | | |
| Social Well-Being(SWB) | 0.934 | 0.935 | 0.740 | 0.38 | 0.29\*\* | 0.38\*\* | 1.00 | |

*Note*: \*\* indicated 0.01 level of significant value

\*Indicates a global rule of thumb of an acceptable level of validity and reliability based on the recommendation of [31–32,51]. α = Cronbach's alpha; AVE = average variance extracted; CR = composite reliability; MSV = maximum shared variance

We established that at least three fit indices (the TLI, CFI, or RMSEA) had to meet predetermined cut-points for a model's fit to be adequate. The cut criteria for changes in model fit indices were 0.10 for CFI and TLI and 0.15 for RMSEA [29,31]. The findings of this study on the Internet Addiction (IA), Emotional Intelligence (EI) and Mental Health (MH) by gender and batch were therefore interpreted using a TLI and CFI threshold of points ΔCFI = 0.02 and of ΔRMSEA = 0.03 for RMSEA [47]. According to model fit comparison indices, the configural MI model of gender on the IA, EI, and MH demonstrated the best model fit, with TLI = 0.941, CFI = 0.951, RMSEA = 0.067, TLI = 0.944, CFI = 0.954, RMSEA = 0.062, and TLI = 0.952, CFI = 0.961, and RMSEA = 0.54, respectively.

Table 5. Fit Indices for Measurement Invariance (Configural, Metric, Scalar, and Residual) Models Across Socio-demographic factors.

| Scales | Groups | Configural | | | Metric | | | Scalar | | | Residual | | |
|---|---|---|---|---|---|---|---|---|---|---|---|---|---|
| | | **TLI** | **CFI** | **RMSEA** | **TLI** | **CFI** | **RMSEA** | **TLI** | **CFI** | **RMSEA** | **TLI** | **CFI** | **RMSEA** |
| Internet Addiction | Gender | 0.941 | 0.951 | 0.067 | 0.945 | 0.952 | 0.065 | 0.949 | 0.952 | 0.063 | 0.943 | 0.943 | .066 |
| | Batch (Years of Study) | 0.923 | 0.936 | 0.064 | 0.926 | 0.934 | 0.063 | 0.928 | 0.930 | 0.062 | 0.912 | 0.908 | .064 |
| Emotional Intelligence | Gender | 0.944 | 0.954 | 0.062 | 0.948 | 0.955 | 0.060 | 0.949 | 0.952 | 0.059 | 0.951 | 0.954 | .058 |
| | Batch (Years of Study) | 0.930 | 0.943 | 0.057 | 0.935 | 0.942 | 0.056 | 0.940 | 0.941 | 0.053 | 0.931 | 0.927 | .057 |
| Mental Health | Gender | 0.952 | 0.961 | 0.054 | 0.957 | 0.962 | 0.051 | 0.950 | 0.952 | 0.055 | 0.955 | 0.953 | .052 |
| | Batch (Years of Study) | 0.940 | 0.951 | 0.049 | 0.939 | 0.946 | 0.049 | 0.940 | 0.940 | 0.049 | 0.945 | 0.940 | .047 |

*Note*: TLI = Tucker Lewis index, CFI = comparative fit index, RMSEA = root mean error square of approximation

For the IA, ΔTLI = −0.004, ΔCFI = −0.001, and RMSEA = 0.002, for EI, ΔTLI = −0.005, ΔCFI = 0.001, and ΔRMSEA = 0.001, for MH, and for MH, ΔTLI = −0.005, ΔCFI = −0.001, and ΔRMSEA = 0.003 were the best fits into the metric invariance model in the data. Additionally, we assessed the residual invariance due to scalar invariance by gender as well as assessing scalar invariance from metric invariance. The results demonstrated that the model satisfactorily fits the data in terms of metric, scalar and residual invariance for the IA, with ΔTLI = −0.004, 0.006, ΔCFI = 0.000, 0.009, and ΔRMSEA = 0.002, −0.003, respectively. The EIQ scores across gender were TLI = −0.004, −0.001, −0.002), ΔCFI = −0.001, 0.003, −0.002, and ΔRMSEA = −0.002, 0.000, 0.001 in metric, scalar and residual invariance, respectively. In terms of metric, scalar and residual invariance across gender, the following values were fitted for MH: ΔTLI = −0.005, 0.007, −0.005; ΔCFI = −0.005, −0.010, −0.001, and RMSEA = −0.003; −0.004 0.003, respectively.

Regarding the batch (years of study), the configural MI demonstrated an acceptable model fit to the data for the three major constructs (the IA, EIQ, and MH) (see Table 5). For the IA, the model comparison test by batch (configural vs. metric; metric vs. scalar, scalar vs. residual) provided metric, scalar, and RMI results, respectively: ΔTLI (−0.005, −0.005, 0.009), ΔCFI (0.001, 0.001, 0.014), and ΔRMSEA = 0.001, 0.003, −0.004), respectively. The metric invariance values for ΔTLI (−0.005, −0.003), ΔCFI 0.001, 0.002), and ΔRMSEA (0.001, 0.001), respectively, indicated that the overall model fit for EI and MH was sufficient. For EI and MH, values for scalar and residual MI had good model fits across batches (ΔTLI = −0.005, 0.001; ΔCFI = 0.009, 0.006; and ΔRMSEA = 0.003, 0.000; respectively). As a result, we can infer that the three primary constructs are equivalent regardless of gender, and batch based on the conventional rule of [29,31].

### 3.1.6. Measurement and structural model.

**Model 1:** Confirmatory Factor Analysis of Internet Addiction Scale (see Fig 4).

**Model 2:** Confirmatory Factor Analysis of Emotional Intelligence Scale (see Fig 5).

**Model 3**: Confirmatory Factor Analysis of Keyes' Mental Health Continuum-Short Form (see Fig 6).

**Model 4**: Internet Addiction → EIQ → Mental Health (see Fig 7)

**Model 5**: Internet Craving, Internet Compulsive Disorder, Addictive Behaviour and Internet Obsession) → EIQ → MH (see Fig 8).

The measurement models (M4 and M5) consisted of three latent constructs and 11 indicators. Specifically, the Internet Addiction Scale (IAS) had four indicators (internet craving, internet compulsive disorder, addictive behaviour and internet obsession), the Emotional Intelligence Scale (EIS-16) had four indicators (self-emotion appraisal; others' emotion appraisal, use of emotions, and regulation of emotions) and Mental Health Continuum-Short Form (MHC-SF) had three indicators (emotional well-being, psychological well-being and social well-being). The measurement model demonstrated good fit based on the confirmatory factor analysis (CFA) results (see Table 6). For the IAS, the model fit indices were $\chi2(113) = 358.10$, $\chi2/df = 3.17$, TLI = 0.953, CFI = 0.961, and RMSEA = 0.081 (95% CI = 0.075, 0.095) (See Fig 4A). The Wong Law Emotional Intelligence Scale (WLEIS-S) also showed acceptable model fit with $\chi2(246) = 274.51$, $\chi2/df = 2.80$, TLI = 0.956, CFI = 0.964, and RMSEA = 0.077 (95% CI = 0.067, 0.088) (See Fig 4B). The MHC-SF exhibited excellent model fit with $\chi2(74) = 171.61$, $\chi2/df = 2.32$, TLI = 0.963, CFI = 0.970, and RMSEA = 0.066 (95% CI = 0.053, 0.073) (See Fig 4C). Overall, the measurement model for all scales demonstrated good fit to the data with $\chi2(1020) = 2219$, $p < .001$, $\chi2/df = 2.12$, TLI = 0.943, CFI = 0.946, and RMSEA = .046

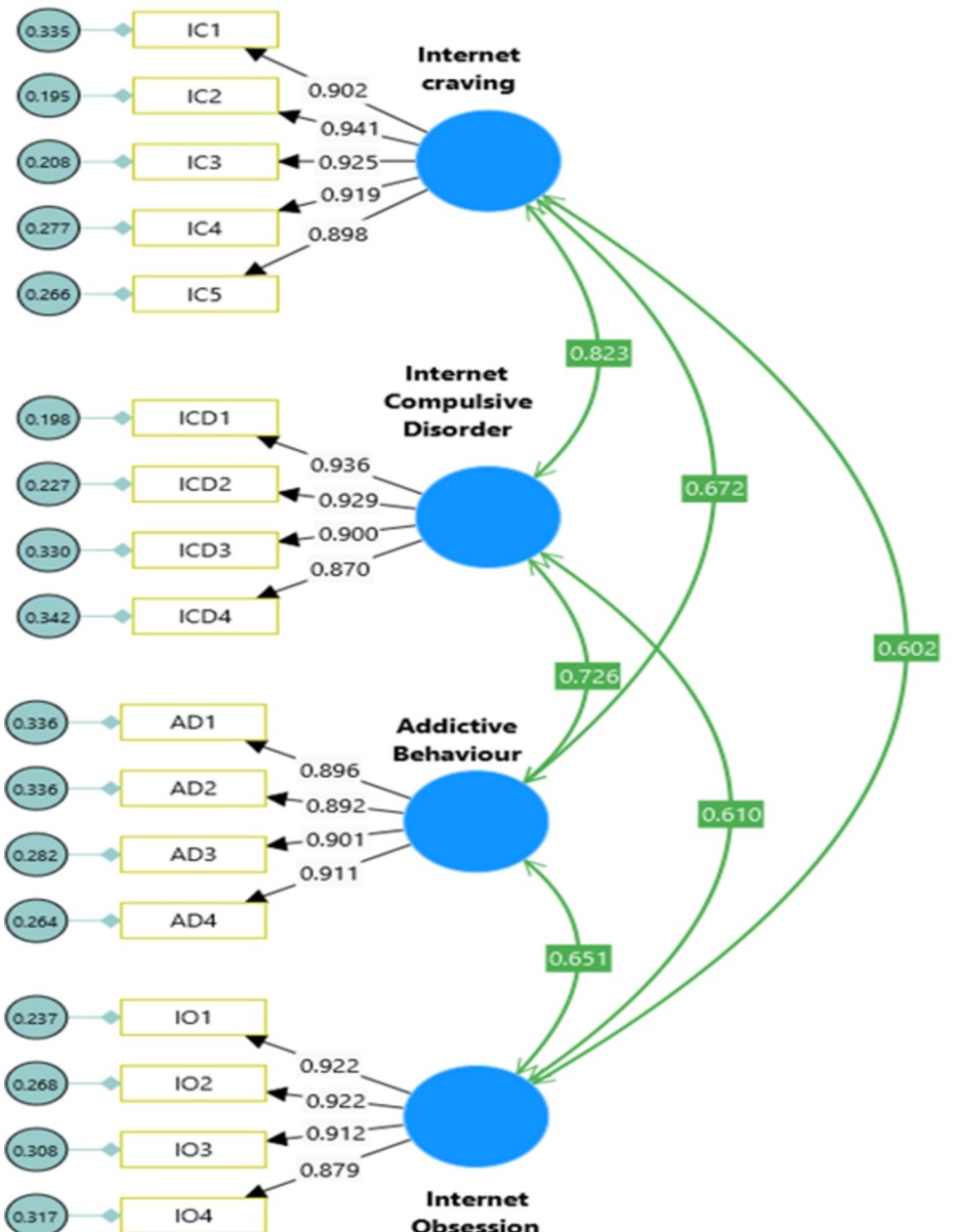

**Fig 4. Confirmatory Factor Analysis model of Internet Addiction Scale.**

(95% CI = 0.044, 0.049), indicating that the latent variables were accurately represented by their indicators. The structural model, which evaluated the relationships between the constructs, also showed good fit to the data with $\chi2(1020) = 2219$, $p < .001$, $\chi2/df = 2.12$, TLI = 0.943, CFI = 0.946, and RMSEA = .046 (95% CI = 0.044, 0.049) (see Table 5). All factor loadings were significant and range in between 0.78 to .92, $p = 0.001$), indicating that the indicators effectively captured the underlying latent variables. In summary, the measurement model and structural model both exhibited good fit to the data (see Table 6), indicating that the indicators accurately represented the latent constructs and the relationships between the constructs were well-supported by the data.

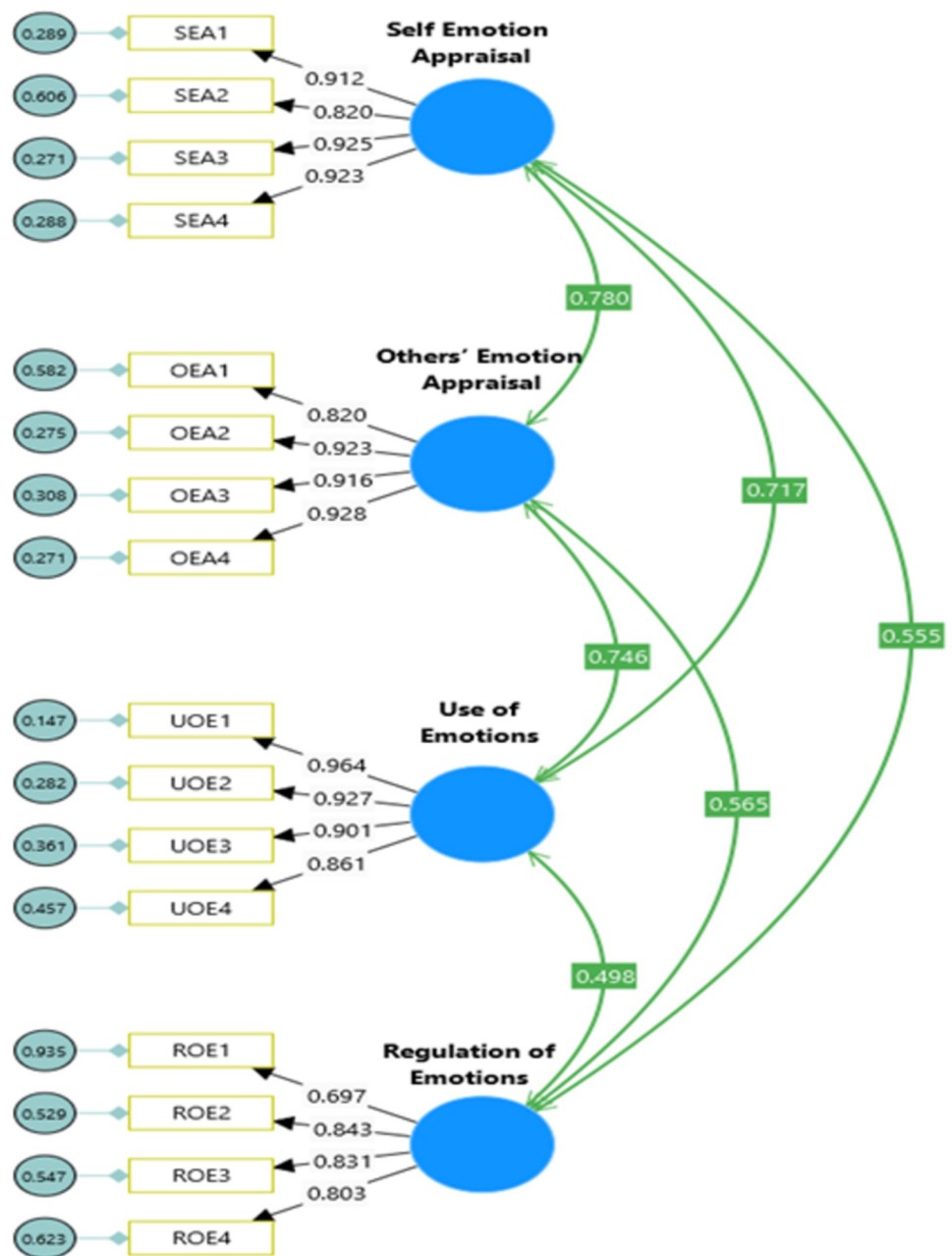

**Fig 5. Confirmatory Factor Analysis model of Emotional Intelligence Scale.**

**3.1.7. Mediation testing using SEM.** The present study employed structural equation modeling (SEM) using the bootstrapping method with latent variables to examine the mediating effects of Emotional intelligence (EIQ) in between internet addiction and mental health was used. Therefore, path analysis was used to examine a mediation model using point estimates and a 95% bootstrap confidence interval for the parameters. The outcome (dependent) variable was mental health, while the predictor (independent) variables were internet addiction and EIQ. The standardized coefficients and 95% confidence intervals obtained using the bootstrap method for the structural model are presented in Table 7 and Figs 7 and 8.

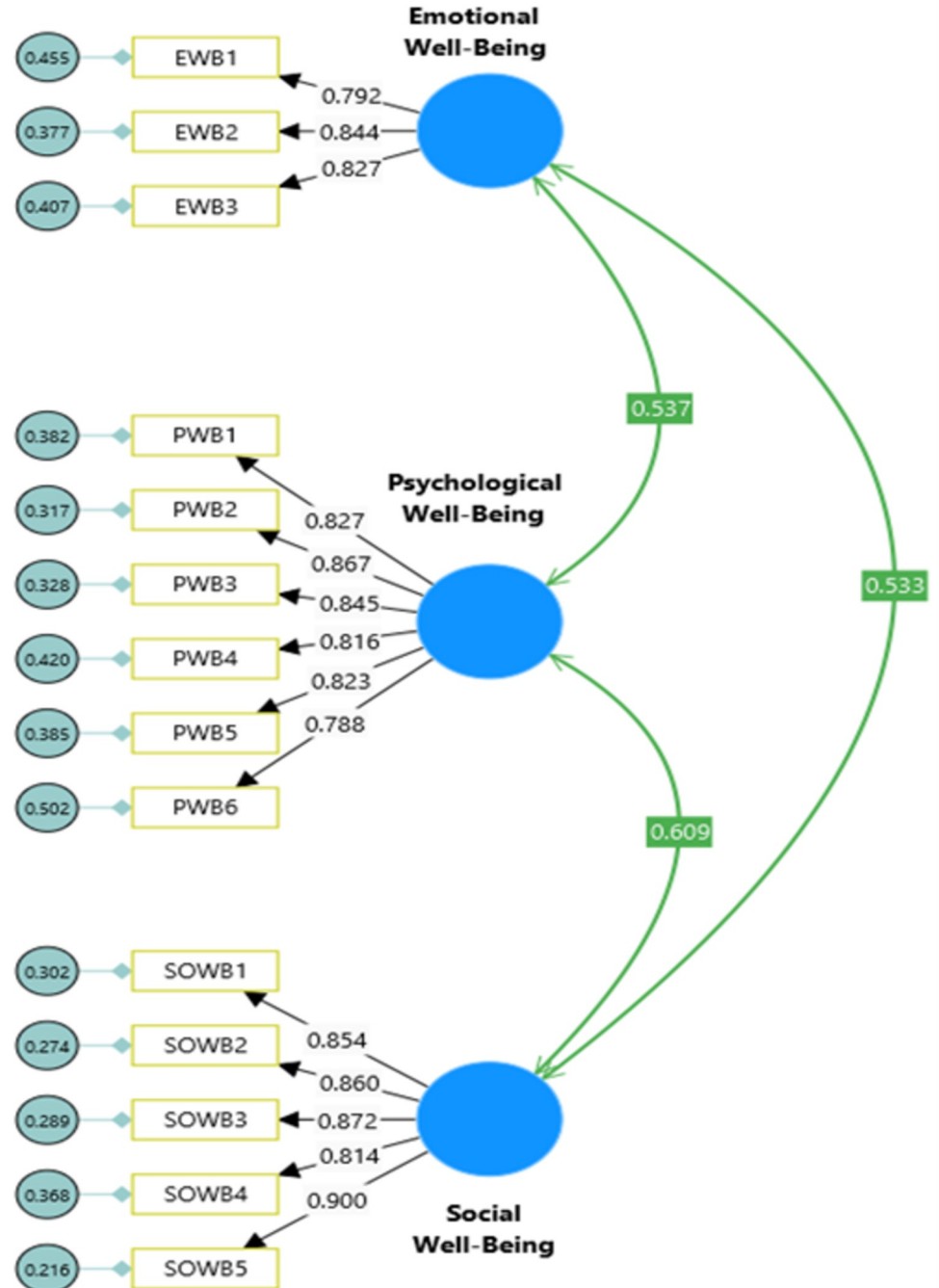

**Fig 6. Confirmatory Factor Analysis model of Mental Health Continuum-Short Form (MHC-SF).**

The results indicated that in the tested model, internet addiction had a direct negative effect on EI (β = −0.180, 95%CI [−0.257, −0.103], p = 0.001) and mental health (β = −0.204, 95%CI [−0.273, −0.134], p = 0.001), supported hypotheses 3 of the findings. However, the study found that EI had a significant and positive direct effect on mental health (β = 0.494, 95%CI [0.390, 0.589], p = 0.001). This finding supported hypotheses 4, Consistent with Hypothesis 5, the

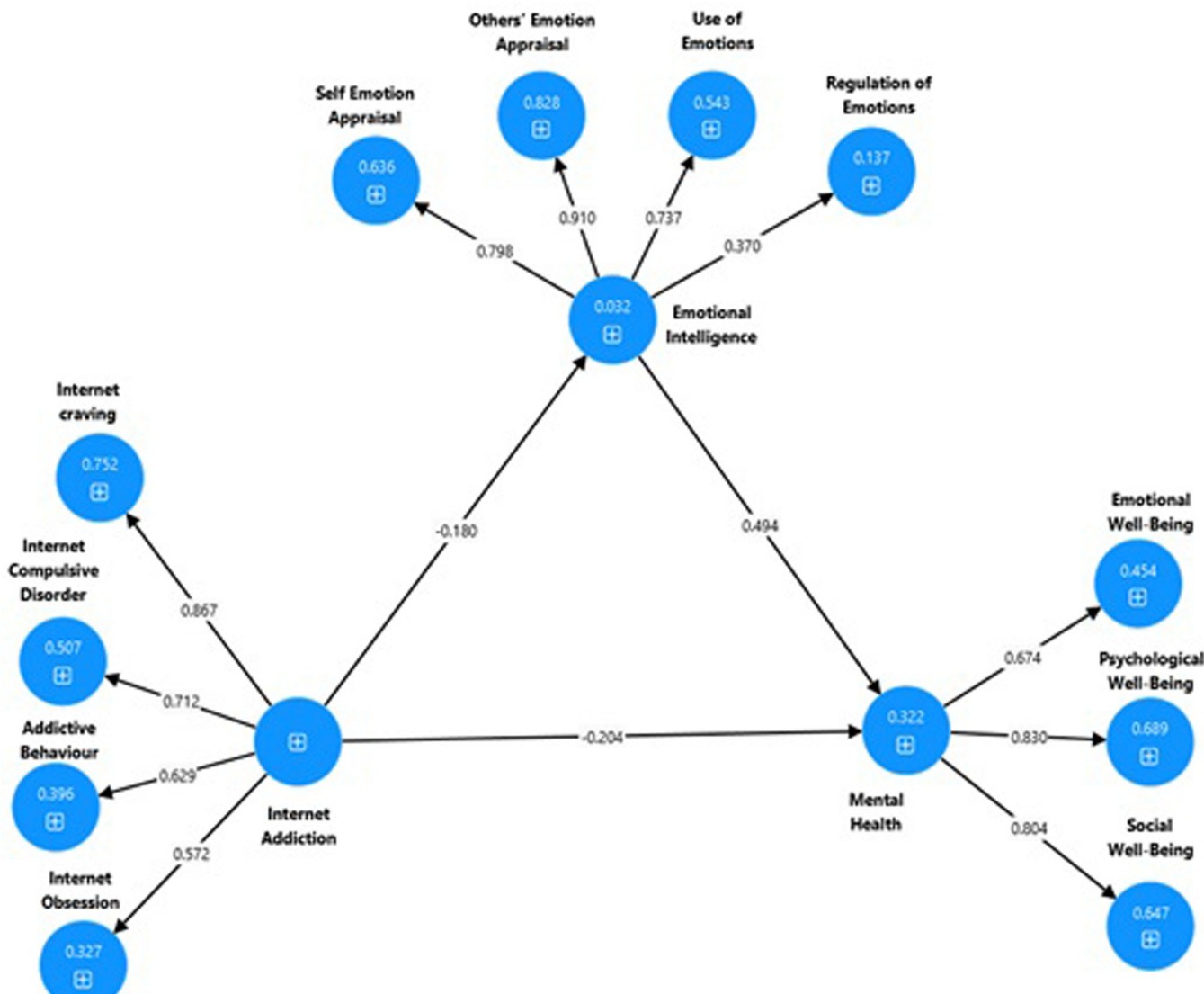

**Fig 7. Output of the mediation model to explain the association between Internet Addiction, Emotional Intelligence and Mental Health.**

study revealed that EI significantly and fully mediated the relationship between internet addiction and mental health (β = –0.089, 95%CI [–0.136, –0.049], p = 0.001).

As displayed iIn Fig 8, we also observed the tested model, which aimed to investigate whether the four dimensions of internet addiction (*Internet craving*, *Internet compulsive disorder*, *Addictive behaviour and Internet obsession*) have an impact on Mental Health (MH) through the mediating effect of Emotional Intelligence. The results revealed several significant relationships. Firstly, we found that Internet carving had a direct negative effect on EIQ, as indicated in Table 7. This suggests that students who possess a high level of internet craving tend to exhibit lower level of emotional intelligence. However, Addictive Behavior positively influence and had a direct significant effect on mental health. This needs further exploration. Secondly, Internet obsession was found to have a negative direct effect on MH, implying that students with a lower level of Internet Obsession are more likely to demonstrate a higher level of mental health.

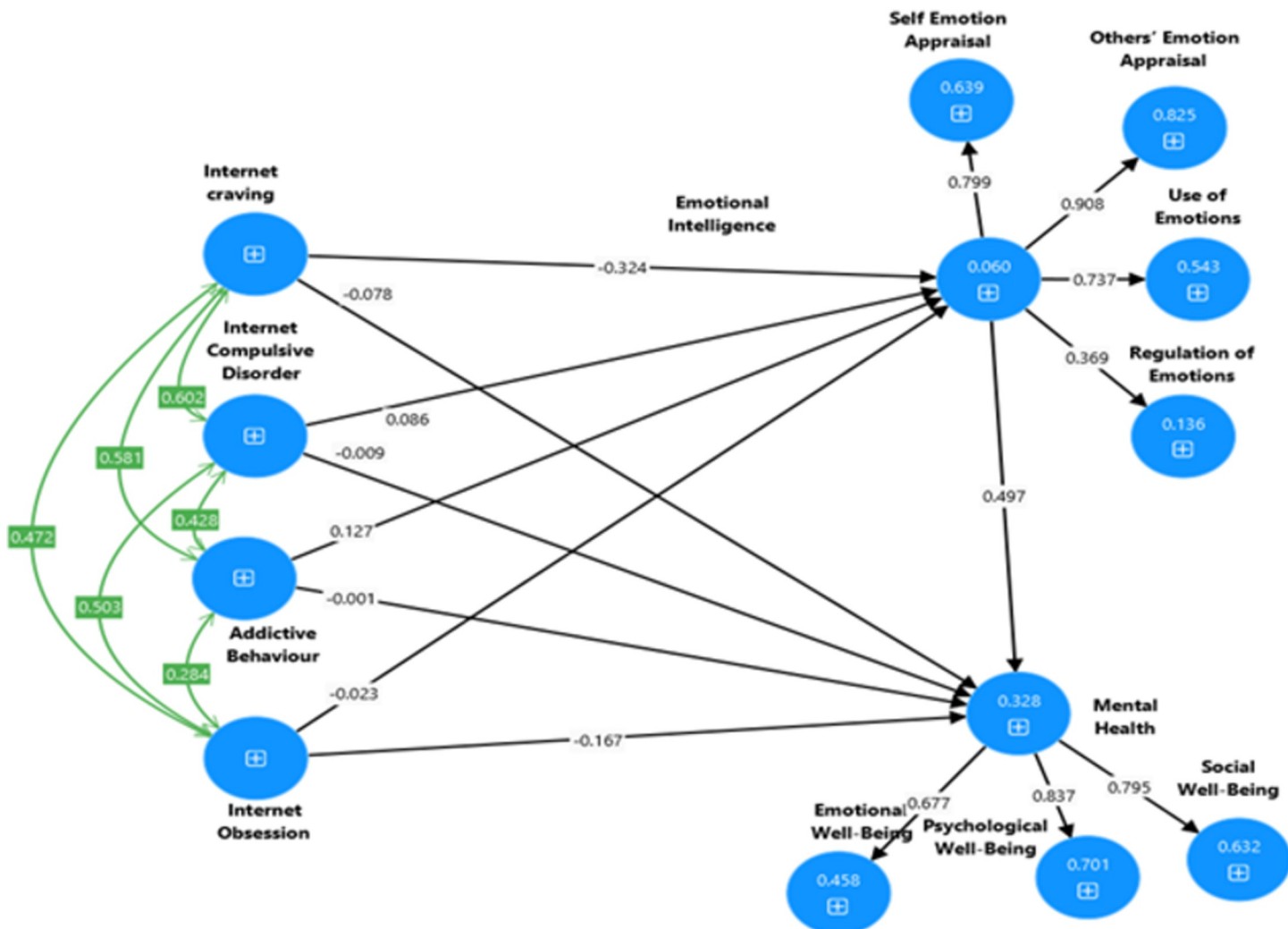

**Fig 8. Output of the mediation model to explain the association between dimensions of Internet Addiction, Emotional Intelligence and Mental Health.**

**Table 6. Confirmatory Factor Analysis of Measurement and the Structural Models of the Constructs.**

| Models | Variables of the Study (N = 300) | Fitness of Indices using Confirmatory Factorial Analysis of the Variables | | | | | |
|---|---|---|---|---|---|---|---|
| | | $\chi^2$ | $\chi^2$/df | TLI | CFI | SRMR | RMSEA |
| Model 1 | Internet Addiction | 358.10 (113) | 3.17 | 0.953 | 0.961 | 0.037 | 0.085 |
| Model 2 | Emotional Intelligence | 274.51 (98) | 2.80 | 0.956 | 0.964 | 0.046 | 0.077 |
| Model 3 | Mental health | 171.61 (74) | 2.32 | 0.963 | 0.970 | 0.034 | 0.066 |
| **Measurement and Structural Models (N = 550)** | | | | | | | |
| Model 4 | Measurement Model | 2219.15 (1020) | 2.12 | 0.943 | 0.947 | 0.047 | 0.046 |
| | Structural Model | 2219.15 (1020) | 2.12 | 0.943 | 0.947 | 0.046 | 0.046 |
| Model 5 | Measurement Model | 2176.31 (1012) | 2.15 | 0.944 | 0.948 | 0.056 | 0.046 |
| | Structural model | 218.27(1012) | 2.15 | 0.944 | 0.947 | 0.056 | 0.046 |
| | Rule of Thumb | | >5 | >0.90 | >0.90 | >0.08 | >1.00 |

*Note*: $\chi^2$ = chi-squared, df = degrees of freedom, TLI = Tucker Lewis index, CFI = comparative fit index, RMSEA = root mean error square of approximation.

**Table 7. Estimated effects of predictors on mental health, including both direct and indirect effects using a 95% biased corrected confidence interval (N = 550).**

| Predictors | Outcome Variables | | Bootstrap 95% CI | | |
|---|---|---|---|---|---|
| | | Beta | LBC | UBC | p-Value |
| Standardized Direct Effect | | | | | |
| Interenet addcition | EIQ | −0.180 | −0.257 | −0.103 | 0.001 |
| Internet addciton | Mental Health | −0.204 | −0.273 | −0.134 | 0.002 |
| Emotional Intelligence (EIQ) | Mental Health | 0.494 | 0.390 | 0.589 | 0.002 |
| Internet Craving | EIQ | −0.324 | −0.196 | −0.043 | 0.002 |
| Internet Craving | Mental Health | −0.078 | −0.188 | −0.080 | 0.256 |
| Internet Compulsive Disorder | EIQ | 0.086 | −0.009 | 0.192 | 0.141 |
| Internet Compulsive Disorder | Mental Health | −0.009 | −0.151 | −0.118 | 0.284 |
| Addictive Behaviour | EIQ | 0.127 | 0.031 | 0.218 | 0.024 |
| Addictive Behaviour | Mental Health | −0.001 | −0.096 | −0.095 | 0.982 |
| Internet Obsession | EIQ | −0.023 | −0.104 | 0.053 | 0.609 |
| Internet Obsession | Mental Health | −0.167 | −0.260 | −0.009 | 0.009 |
| Standardized Indirect Effect | | | | | |
| Interenet addcition → EI → | Mental Health (Fig 7 | −0.089 | −0.136 | −0.049 | 0.001 |
| Internet Craving → EI → | Mental Health (Fig 8) | −0.163 | −0.230 | −0.102 | 0.001 |
| Internet compulsive disorder→ EI → | Mental Health (Fig 8) | −0.043 | −0.002 | −0.103 | 0.120 |
| Addictive Behaviour→ EI → | Mental Health (Fig 8) | 0.053 | 0.018 | −0.115 | 0.129 |
| Internet Obsession→ EI → | Mental Health (Fig 8) | −0.011 | −0.054 | 0.026 | 0.571 |

*Note*: CI = confidence interval, LBC = lower bound, UBC = upper bound

Furthermore, the study found that a substantial and positive indirect effect Internet Craving on students' mental health. In contrast, Addictive Behavior indirectly and positively influenced MH, as evidenced by the findings presented in Table 7 and Fig 8. Our data partially confirmed RH5, highlighting the significant relationships between the dimensions of IA, and MH among university students.

## 4. Discussion

This study investigated two important issues: the psychometric properties of the three measures and the mediation analysis. Following the correlation among the constructs in the first research hypothesis, the second hypotheses aimed at testing the psychometric properties of the three measures namely, the Wong Law Emotional Intelligence Scale (WLEIS-S), Internet Addiction Scale (IAS) and Keyes' Mental Health Continuum-Short Form (MHC-SF) of Amharic version in Ethiopian context. Hence, the Ethiopian Amharic version of IAS, WLEIS-S, MHC-SF were checked Cronbach alpha's and composite reliability, convergent, discriminant and construct validity and proved the three measures were reliable and valid. The AVE is greater than 0.05, and AVE was less than MSV, squared correlation of the sub constructs which we used with a sample of Ethiopian university students. In the second goal of the study as seen in the hypothesis 5 (see Fig 7) that internet addiction (IA) predicts mental health (MH) as the dependent variable, with Emotional Intelligence (EIQ) serving as a mediating variable. The study also examined the relationship between IA and MH, with EI as a mediating variable. The results showed a negative correlation between IA, EIQ and MH, indicating that higher levels of IA were associated with lower levels of EIQ and MH. However, EIQ and MH were positively correlated, suggesting that higher levels of EI were associated with better MH. The negative prediction of internet addiction on both EIQ and mental health highlights the

detrimental effects of excessive internet use on the mental well-being of university students. Internet addiction is a growing concern in today's digitally connected world, with students being particularly vulnerable due to the extensive use of online platforms for study, communication, and entertainment. In addition, Internet Craving directly predicted EIQ and indirectly and negatively predicted mental health through EIQ. However, Addictive behavior directly and positively predicted EIQ and indirectly predicted MH. This finding needs further exploration cross culturally. These findings are consistent with several studies suggesting a link between Emotional Intelligence (EIQ) and Mental Health (MH). The hypothesis supported by the previous scientific evidence that internet addiction has a negative impact on mental health, showing a strong relationship between them [3,6,10–14,17].

Additionally, [57] conducted a study on "gender-linked personality and mental health: The role of trait emotional intelligence" and found a significant negative relationship between Emotional Intelligence and mental health. Other studies have also found that higher levels of internet addiction are associated with lower Emotional Intelligence and poorer mental health. For example, [6,57–58], found that higher levels of internet addiction were linked to lower Emotional Intelligence and poorer mental health. Similarly, [6,59] found that internet addiction was associated with lower moral values and psychological well-being, while [49,52–53] discovered a negative influence on emotional intelligence.

Furthermore, [26–28] found that addiction to social networking sites (SNS) and higher levels of internet addiction were connected to higher levels of moral disengagement and poorer mental health. These findings have significant implications for interventions and preventive measures aimed at addressing internet addiction and promoting mental health among university students. Efforts should be directed towards raising awareness about the potential risks associated with excessive internet use and providing strategies for developing healthy digital habits. In addition, interventions should focus on enhancing emotional intelligence by promoting activities that facilitate the appraisal of others' emotions, regulation of emotions, self-appraisal of emotions, and effective use of emotions. This finding is consistent with previous research that highlights the protective role of these factors in promoting mental well-being [6,60–63]. It suggests that interventions aimed at improving emotional intelligence can positively impact mental health, particularly among university students who often face various stressors and challenges during their academic journey. University counseling services and educational programs can play a vital role in supporting students in developing coping skills, managing stress, and maintaining a healthy balance between online and offline activities. By providing resources and guidance, these services can contribute to the overall well-being of students and help them navigate the demands of both their online and offline lives.

The second main goal of the study was the mediating role of Emotional Intelligence in the relationship between internet addiction and mental health. There are several possible explanations for the observed mediation effect of Emotional Intelligence. *First*, excessive internet use can lead to diminished engagement in offline activities, such as social interactions, physical exercise, and face-to-face communication, which are crucial for developing and maintaining better emotional intelligence capabilities. *Second*, internet addiction may contribute to feelings of isolation, loneliness, and decreased self-esteem, which can further deplete Emotional Intelligence and negatively impact mental health. *Third*, excessive internet use might disrupt sleep patterns, leading to fatigue and impaired cognitive functioning, which can directly influence Emotional Intelligence and mental health as well as indirectly affect mental health of university students.

The study suggests that positive interventions focusing on using Emotional Intelligence, along with the development of healthy digital system, improves the mental health of university students and their daily functioning. These interventions may lead to personal and

organizational development and growth. This line of reasoning is supported by prior studies that have found Emotional Intelligence to be the best predictors of mental health and positive outcomes on students' life, reducing stress, and fostering healthy digital functioning [10,13,17,61–63]. Moreover, Emotional Intelligence has been identified as a preventive resource that can be used to improve mental health and lower internet addiction, and empirical evidence has shown a negative relationship with internet addiction and a positive relationship with mental health, which supports the hypotheses of this study.

It is important to note that this study focused specifically on university students, and further research is needed to generalize these findings to other populations. Additionally, the study relied on self-report measures, which may be subject to biases and limitations. Future research could employ longitudinal designs to examine the causal relationships between Internet Addiction, Emotional Intelligence, and Mental Health, as well as explore potential moderators and other mediating factors that may contribute to the observed associations.

In conclusion, this study provides valuable insights into the impact of internet addiction on mental health in university students. The findings underscore the importance of addressing internet addiction and promoting Emotional Intelligence as key factors in enhancing mental well-being among this population. These findings contribute to understanding the relationships between IA, EI, and MH and highlight the importance of positive interventions and the better mental health and healthy internet usage and positive EI for promoting mental well-being among university students. By recognizing and addressing these issues, universities and relevant stakeholders can contribute to the overall holistic development and mental health of their students.

## 5. Conclusion

The purpose of the current study was to test the psychometric suitability of the three measures namely IAS, EIS-16 and MHC-SF and examine the mediation role of emotional intelligence (EIQ) between internet addiction (IA) and Mental Health (MH). The results showed that IA had a significant and negative direct effect on Emotional Intelligence and MH. Additionally, Emotional Intelligence had a positive direct impact on MH and fully mediated the relationship between IA and MH. This indicates that higher levels of Internet Addiction among university students are associated with lower levels of Emotional Intelligence, which encompasses positive emotional resources such as others' emotion appraisal, regulation of emotions, self-emotion appraisal and use of emotions. Furthermore, lower Emotional Intelligence is associated with poorer mental health outcomes.

The finding that Emotional Intelligence fully mediates the relationship between internet addiction and mental health suggests that the detrimental effect of internet addiction on mental health is primarily driven by its impact on Emotional Intelligence resources. However, emotional intelligence partially mediated the relationship among dimensions of IA and mental health. In other words, when university students have higher levels of internet addiction, it leads to lower Emotional Intelligence, which, in turn, contributes to poorer mental health outcomes. These results emphasize the importance of addressing internet addiction in university students, as it not only directly affects mental health but also indirectly influences mental health through its impact on Emotional Intelligence. These findings underscore the importance of implementing interventions and preventive measures that target both internet addiction and the promotion of Emotional Intelligence resources among university students. By reducing internet addiction and fostering Emotional Intelligence, we can enhance mental well-being and mitigate the negative effects of internet addiction on mental health.

Practitioners and researchers should consider incorporating strategies that aim to reduce excessive internet use and enhance Emotional Intelligence skills in their programs and interventions. This may involve providing educational resources, counseling services, and workshops that focus on improving Emotional Intelligence competencies such as self-awareness, self-regulation, empathy, and relationship management. By addressing both internet addiction and Emotional Intelligence, we can support the mental well-being of university students and promote healthier online and offline behaviors.

Overall, the findings emphasize the need for a comprehensive approach to addressing the interplay between internet addiction, Emotional Intelligence, and mental health among university students. By targeting these factors simultaneously, we can develop effective strategies to enhance mental health outcomes and promote overall well-being in this population.

# 6. Implications and future research suggestions

## 6.1. Implications for the researchers

The psychometric properties of the measures (Wong Law Emotional Intelligence Scale, Internet Addiction Scale, and Keyes' Mental Health Continuum-Short Form) have been established in the Ethiopian Amharic context. Researchers can confidently use these measures in future studies involving university students. The study confirms the negative impact of internet addiction on both Emotional Intelligence (EIQ) and mental health (MH) among university students. This highlights the need for interventions and preventive measures to address internet addiction and promote mental well-being.

The study highlights the mediating role of Emotional Intelligence between internet addiction (dimensions and general factor) and mental health. Researchers can explore further the underlying mechanisms and processes through which Emotional Intelligence influences the relationship between internet addiction and mental health.

The study emphasizes the importance of raising awareness about the potential risks associated with excessive internet use and providing strategies for developing healthy digital habits among university students. Researchers can collaborate with educational institutions to design and implement interventions targeting digital well-being and emotional intelligence enhancement.

## 6.2. Suggestions for future research

**Generalization.** While this study focused on university students, future research should aim to generalize these findings to other populations, such as different age groups or socio-cultural contexts. This would provide a more comprehensive understanding of the relationships between internet addiction, Emotional Intelligence, and mental health across diverse populations.

**Longitudinal Studies.** Employing longitudinal designs would allow researchers to explore the causal relationships between internet addiction, Emotional Intelligence, and mental health. Longitudinal studies would provide insights into the temporal dynamics and potential directionality of these relationships.

**Moderators and Mediators.** Future research could investigate potential moderators and other mediating factors that may influence the associations between internet addiction, personality traits, positive psychological capital, emotional intelligence, mindfulness and mental health [6,10,13,17,63]. For example, variables such as social support, social capital, self-esteem, or coping strategies could be examined to better understand the complex interplay between these constructs.

**Intervention Studies.** Conducting intervention studies that focus on improving Emotional Intelligence and promoting healthy digital habits among university students would provide valuable insights into effective strategies for enhancing mental well-being. These studies can evaluate the impact of interventions on reducing internet addiction and improving mental health outcomes.

**Comparative Studies.** Comparing the impact of different types of online activities (e.g., social media use, gaming, browsing) on Emotional Intelligence and mental health could help identify specific areas of concern and inform targeted interventions.

**Use of Objective Measures.** While self-report measures are commonly used, future research could consider incorporating objective measures (e.g., behavioral observations, physiological indicators) to complement self-report data and provide a more comprehensive understanding of the relationships between internet addiction, Emotional Intelligence, and mental health.

By addressing these research implications and pursuing future studies based on the data, researchers can further advance the knowledge in this field, contribute to the development of effective interventions, and promote the well-being of university students in the digital age.

## Supporting information

**S1 File. All confirmatory Factor Analysis Results and Structural Equation Modelling outputs.**
(ZIP)

## Author Contributions

**Conceptualization:** Girum Tareke Zewude, Derib Gosim, Seid Dawed, Getachew Wassie Tessema, Amogne Asfaw Eshetu.

**Data curation:** Tilaye Nega.

**Formal analysis:** Girum Tareke Zewude.

**Investigation:** Derib Gosim, Tilaye Nega, Getachew Wassie Tessema.

**Methodology:** Girum Tareke Zewude.

**Resources:** Derib Gosim, Seid Dawed, Amogne Asfaw Eshetu.

**Validation:** Girum Tareke Zewude.

**Writing – original draft:** Girum Tareke Zewude, Seid Dawed, Getachew Wassie Tessema.

**Writing – review & editing:** Girum Tareke Zewude, Derib Gosim, Tilaye Nega, Amogne Asfaw Eshetu.

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
