## [Decision Letter · Decision Letter 0]

28 May 2024

PDIG-D-24-00085

The Impact of Internet Addiction on Mental Health: Exploring the Mediating Effects of Emotional Intelligence in University Students

PLOS Digital Health

Dear Dr. Zewude,

Thank you for submitting your manuscript to PLOS Digital Health. After careful consideration, we feel that it has merit but does not fully meet PLOS Digital Health's publication criteria as it currently stands. Therefore, we invite you to submit a revised version of the manuscript that addresses the points raised during the review process.

Please submit your revised manuscript within 60 days Jul 27 2024 11:59PM. If you will need more time than this to complete your revisions, please reply to this message or contact the journal office at digitalhealth@plos.org. Please include the following items when submitting your revised manuscript:

We look forward to receiving your revised manuscript.

Kind regards,

Calvin Or, PhD

Section Editor

PLOS Digital Health

Journal Requirements:

Additional Editor Comments (if provided):

Reviewers' comments:

Reviewer's Responses to Questions

**Comments to the Author**

1. Does this manuscript meet PLOS Digital Health’s publication criteria? Is the manuscript technically sound, and do the data support the conclusions? The manuscript must describe methodologically and ethically rigorous research with conclusions that are appropriately drawn based on the data presented.

Reviewer #1: Yes

Reviewer #2: Yes

2. Has the statistical analysis been performed appropriately and rigorously?

Reviewer #1: Yes

Reviewer #2: Yes

3. Have the authors made all data underlying the findings in their manuscript fully available (please refer to the Data Availability Statement at the start of the manuscript PDF file)?

Reviewer #1: No

Reviewer #2: Yes

4. Is the manuscript presented in an intelligible fashion and written in standard English?

Reviewer #1: Yes

Reviewer #2: Yes

5. Review Comments to the Author

Reviewer #1: I have comprehensive review of your paper. I have provided constructive feedback aimed at enhancing the quality and impact of your article. I encourage you to consider these suggestions to improve clarity and overall effectiveness. Upon implementing revisions, I am eager to review the updated version. Your commitment to refining your work is admirable, and I am eager to witness the evolution of your article. Best regards.

Abstract

1. it could benefit from specifying the methodology briefly, such as mentioning the use of structural equation modeling (SEM) explicitly.

2. Ensure that the results section provides a clear and concise summary of the findings without interpretation. It should focus on presenting the statistical outcomes of the analyses conducted.

3. Consider including effect sizes or confidence intervals to complement the significance testing and provide a more comprehensive understanding of the results.

4. Expand on the practical implications for practitioners and researchers. How can the findings inform the development of interventions or policies to address internet addiction and promote mental health in university settings?

introduction:

• In the introduction, it is imperative to commence with a delineation of the study group, followed by an exposition of the dependent variables, and culminating with an elucidation of the independent variables. Seamless cohesion should characterize the transition between paragraphs, akin to the interlocking links of a chain. The background of the research ought to succinctly delineate the existing lacunae and unresolved queries, thereby fostering clarity for the reader. In the concluding segment, it is incumbent upon the author to articulate the persisting issues and research lacunae, alongside both the direct and indirect implications of the study. Emphasis should be placed on elucidating how the findings of this research endeavor can contribute to the resolution of ambiguities.

• it could benefit from a clearer structure to guide the reader through the main points. Consider dividing the introduction into subsections to address different aspects of the topic sequentially.

• Ensure smooth transitions between paragraphs to maintain coherence and flow in the narrative.

• provide a more focused and concise review of the internet addiction, emotional intelligence, and mental health. Try to highlight key findings and theories relevant to the research objectives more explicitly.

• To enhance the quality of your introduction, consider incorporating the following references:

1. https://brieflands.com/articles/ijpbs-9254

2. https://jpcp.uswr.ac.ir/browse.php?a_id=862&sid=1&slc_lang=en&html=1

3. https://www.behavsci.ir/article_192588.html

4. http://jpcp.uswr.ac.ir/browse.php?a_id=862&sid=1&slc_lang=en&html=1

5. https://link.springer.com/article/10.1007/s11469-021-00617-9

6. https://link.springer.com/article/10.1007/s12646-023-00713-x

• The introduction briefly mentions the cognitive-behavioral model of internet addiction and the broaden-and-build theory of positive emotions. It would be helpful to provide a more detailed explanation of how these theoretical frameworks inform the study's hypotheses and research design.

• Consider providing a brief explanation of the components of the model (emotional intelligence, internet addiction, mental health) to aid the reader's understanding.

method:

1. please explain how the study size was arrived at.

2. The methodology section needs to provide more detail. In particular, you should provide more information about the data collection and analysis methods you used. Readers need to understand how you collected and analyzed your data in order to evaluate the validity of your results.

3. Ensure that each subsection is clearly labeled and addresses a specific aspect of the research method (e.g., Research Design, Study Setting, Sample and Sampling, Instruments, Statistical Data Analysis, Procedures).

4. Consider providing more information on how the sample size was determined, particularly regarding the power analysis or considerations specific to the chosen statistical methods.

5. Ensure consistency in terminology when describing the psychometric properties of the instruments (e.g., referring to Cronbach's alpha as reliability coefficient consistently).

6. Consider providing more detail on how missing data were handled, particularly regarding the exclusion of participants due to missing information.

Results

• The authors appropriately conducted descriptive statistics and checked for normality, which is crucial for inferential statistics. However, it would be helpful to include information on the sample size and demographic characteristics to contextualize the findings.

• Before presenting the tables, the demographic characteristics of the participants should be fully expressed in a paragraph.

• It would be helpful to include a clear explanation of the theoretical rationale behind testing mediation and how the findings contribute to existing literature.

• The inclusion of tables and figures enhances the presentation of results. Ensure that all tables and figures are appropriately labeled and referred to in the text.

discussion:

1. The discussion makes several claims regarding the relationship between internet addiction, emotional intelligence, and mental health. While these claims are supported by citations, there's room to strengthen the discussion by providing more detailed explanations of how each cited study contributes to the understanding of the topic. Additionally, ensure consistency in citation formatting throughout the discussion.

2. When discussing the study findings, provide a more detailed interpretation of the results. For example, elaborate on why higher levels of internet addiction lead to lower emotional intelligence and poorer mental health outcomes. Providing insights into the underlying mechanisms or pathways can enhance the understanding of the study's implications.

3. Consider expanding on limitations and providing specific suggestions for future research directions. For example, discuss potential strategies for addressing the limitations, such as using objective measures or longitudinal designs.

4. it could benefit from a more detailed exploration of how implications can be translated into real-world interventions or policies. Provide concrete examples of interventions or strategies that universities or relevant stakeholders can implement to address internet addiction and promote emotional intelligence among students.

5. Consider reinforcing the main takeaway messages and emphasizing the importance of addressing internet addiction and promoting emotional intelligence in university settings. Additionally, reiterate the need for a comprehensive approach to addressing mental health among university students.

6. In the limitations section, it is important to identify the factors that have impacted the internal and external credibility of the research, as well as any methodological limitations. Based on these limitations, research suggestions should be proposed, and practical recommendations should be provided. It is crucial to avoid making general suggestions, such as merely explaining the findings based on the results of the hypothesis.

7. Recommendations for future research are an essential aspect of developing new ideas in the field. This article should, therefore, include suggestions for new questions that need to be addressed through future empirical research, and should pose questions and hypotheses that need to be the targets of new studies.

Reviewer #2: The manuscript discusses The Impact of Internet Addiction on Mental Health: Exploring the Mediating Effects of Emotional Intelligence in University Students.

The study aims to examine whether Internet addiction negatively predicts mental health in college students, with emotional intelligence acting as a mediator.

This study sheds light on the results revealed that Internet addiction had a negative and direct effect on emotional intelligence and mental health.

The results highlight the detrimental effects of Internet addiction on mental health, and the crucial mediating role of emotional intelligence.

Authors should check whether the abstract has been written as requested in the journal guidelines.

The introduction and discussion should be strengthened, so I indicate below some elements that authors should improve.

It is important that the authors take into account Psychosocial risk factors of technological addictions in a sample University students: The influence of Emotional (Dys) Regulation, personality traits and Fear of Missing Out on internet addiction.

Since the sample is gender balanced, it would be interesting for the authors to make some mention about Gender differences in internet addiction. It would be very interesting if the authors would address in the introduction From Emotional (Dys)Regulation to Internet Addiction. 

The introduction would be very complete if the authors would address the Relationship between Problematic Smartphone Use, Sleep Quality and Bedtime Procrastination 

Authors should explicitly include, in the introduction, the question to be answered and return to it in the discussion.

The method is well described

In the discussion, it is convenient to take into account, also the suggestions that I have included to strengthen the introduction. As well as the relationship between Problematic Internet Use and Resilience: 

The conclusions are strong with the evidence and arguments presented. The citations and references are appropriate and up to date, but the authors should review the journal guidelines, so that they are as requested in Plos One. The tables are very clear, but it is advisable to check whether they are in the format requested by the journal in which they are to be published.

The authors contribute knowledge to science with this manuscript, being relevant and of interest.

6. PLOS authors have the option to publish the peer review history of their article (what does this mean?). If published, this will include your full peer review and any attached files.

**Do you want your identity to be public for this peer review?** For information about this choice, including consent withdrawal, please see our Privacy Policy.

Reviewer #1: No

Reviewer #2: No

---

## [Decision Letter · Decision Letter 1]

11 Sep 2024

Investigating the Mediating Role of Emotional Intelligence in the Relationship between Internet Addiction and Mental Health among University Students

PDIG-D-24-00085R1

Dear Dr. Zewude,

We are pleased to inform you that your manuscript 'Investigating the Mediating Role of Emotional Intelligence in the Relationship between Internet Addiction and Mental Health among University Students' has been provisionally accepted for publication in PLOS Digital Health.

Best regards,

Calvin Or, PhD

Section Editor

PLOS Digital Health

Reviewer Comments (if any, and for reference):

Reviewer's Responses to Questions

**Comments to the Author**

1. If the authors have adequately addressed your comments raised in a previous round of review and you feel that this manuscript is now acceptable for publication, you may indicate that here to bypass the “Comments to the Author” section, enter your conflict of interest statement in the “Confidential to Editor” section, and submit your "Accept" recommendation.

Reviewer #1: All comments have been addressed

Reviewer #2: (No Response)

2. Does this manuscript meet PLOS Digital Health’s publication criteria? Is the manuscript technically sound, and do the data support the conclusions? The manuscript must describe methodologically and ethically rigorous research with conclusions that are appropriately drawn based on the data presented.

Reviewer #1: Yes

Reviewer #2: (No Response)

3. Has the statistical analysis been performed appropriately and rigorously?

Reviewer #1: I don't know

Reviewer #2: Yes

4. Have the authors made all data underlying the findings in their manuscript fully available (please refer to the Data Availability Statement at the start of the manuscript PDF file)?

Reviewer #1: No

Reviewer #2: Yes

5. Is the manuscript presented in an intelligible fashion and written in standard English?

Reviewer #1: Yes

Reviewer #2: Yes

6. Review Comments to the Author

Reviewer #1: I can see authors revised and the paper is acceptable.

Regardd

Reviewer #2: The authors have addressed all the issues raised.

For my part, the manuscript is complete and can be published.

7. PLOS authors have the option to publish the peer review history of their article (what does this mean?). If published, this will include your full peer review and any attached files.

**Do you want your identity to be public for this peer review?** For information about this choice, including consent withdrawal, please see our Privacy Policy.

Reviewer #1: No

Reviewer #2: No
